# Activation of GPR56, a novel adhesion GPCR, is necessary for nuclear androgen receptor signaling in prostate cells

**Julie Pratibha Singh**[1], **Manisha Dagar**[1], **Gunjan Dagar**[1], **Sudhir Kumar**[2], **Sudhir Rawal**[3], **Ravi Datta Sharma**[4], **Rakesh Kumar Tyagi**[2], **Gargi Bagchi**[1] *

**1** Amity Institute of Biotechnology (AIB), Amity University Haryana, Manesar, Gurugram, India, **2** Special Centre for Molecular Medicine, Jawaharlal Nehru University, New Delhi, India, **3** Rajiv Gandhi Cancer Institute & Research Centre, Rohini, New Delhi, India, **4** Amity Institute of Integrative Sciences and Health (AIISH), Amity University Haryana, Manesar, Gurugram, India

* gbagchi@ggn.amity.edu

**Data Availability Statement:** All relevant data are within the paper and its supporting information files.

## Abstract

The androgen receptor (AR) is activated in patients with castration resistant prostate cancer (CRPC) despite low circulating levels of androgen, suggesting that intracellular signaling pathways and non-androgenic factors may contribute to AR activation. Many G-protein coupled receptors (GPCR) and their ligands are also activated in these cells indicating that they may play a role in development of Prostate Cancer (PCa) and CRPC. Although a cross talk has been suggested between the two pathways, yet, the identity of GPCRs which may play a role in androgen signaling, is not established yet. By using blast analysis of 826 GPCRs, we identified a GPCR, GPCR 205, which exhibited maximum similarity with the ligand binding domain of the AR. We demonstrate that adhesion GPCR 205, also known as GPR56, can be activated by androgens to stimulate the Rho signaling pathway, a pathway that plays an important role in prostate tumor cell metastasis. Testosterone stimulation of GPR56 also activates the cAMP/ Protein kinase A (PKA) pathway, that is necessary for AR signaling. Knocking down the expression of GPR56 using siRNA, disrupts nuclear translocation of AR and transcription of prototypic AR target genes such as PSA. GPR56 expression is higher in all twenty-five prostate tumor patient's samples tested and cells expressing GPR56 exhibit increased proliferation. These findings provide new insights about androgen signaling and identify GPR56 as a possible therapeutic target in advanced prostate cancer patients.

## Introduction

The AR plays a central role in the normal growth of prostate as well as its neoplastic transformation into PCa [1]. The fundamental dependence of prostate tumor on androgen signaling was established when Huggins and Hodges in 1941 demonstrated that orchiectomy in patients could reduce tumor growth [2]. Since then endocrine therapies aiming to minimize AR signaling, either by reducing androgen levels by physical or chemical means or by using AR inhibitors, constitute the primary treatment plan for PCa patients [3]. These measures however offer only a temporary relief as a more aggressive and androgen-independent form of tumor inevitably reappears, that is untreatable [4].

**Funding:** This work was supported by Department of Biotechnology(DBT), India No.BT/Bio-CARe/01/668/2011-12 (2014-2018) (to GB). The funders had no role in study design, data collection and analysis, decision to publish, or preparation of the manuscript.

**Competing interests:** The authors have declared that no competing interests exist.

Analysis of tumor samples from patients with CRPC or xenograft models demonstrate that the reasons for androgen independence include overexpression of AR in many of these tumors, mutations in AR and its alternatively spliced forms [5], increased coactivator expression [6] and activation of AR by molecules such as cytokines or growth factors which cause stimulation of other cellular pathways that indirectly activate the AR [7, 8].

Indeed, the release of circulating and locally synthesized factors acting via cellular receptors cause a switch of the quiescent prostate cells into an activated state causing uncontrolled cellular proliferation [9]. Among the cellular receptors that are implicated in PCa, the most prominent are the G-protein coupled receptors (GPCR) [10]. The cancerous prostate contains elevated levels of enzymes such as kallikrein 2 which control the expression of GPCR ligands [11] and it also produces increased amount of GPCR ligands such as endothelin 1 [12], follicle stimulating hormone and lysophosphatidic acid [13]. Compared to normal prostate higher expression of various GPCRs is commonly observed in the malignant prostate such as FSH receptor [14], bradykinin 1 receptor [15], and Endothelin 1A receptor [12]. These observations taken together strongly indicate that the GPCR system may be in the 'on' state in the malignant prostate contributing directly or indirectly to tumor growth [9].

The classical mechanism of androgen signaling involves binding of androgen to the cytosolic AR, which dimerizes and moves into the nucleus to bind androgen response elements upstream of target genes to modulate gene expression and cause cellular proliferation [16]. This genomic signaling by androgens is mediated in hours or days. However, studies over the last decade has established that androgens also initiate rapid, non-genomic changes in target cells, which may or may not involve the nuclear AR [17]. Non-genomic signaling by androgens include activation of extracellular signal regulated kinase in Sertoli cells [18], phosphatidyl inositol 3-kinase in osteoblasts [19] or extracellular $Ca^{2+}$ in skeletal muscle cells [20]. Androgens also initiate rapid activation of protein kinase A (PKA) in prostate cells, via a GPCR, and androgen mediated PKA activation is necessary for normal functioning of AR [17]. In fact, the activation of PKA by forskolin or other non-androgenic agents can also lead to stimulation of AR and cause transcription of androgen target genes [21]. Activation of these non-genomic pathways appear to play a key role in development of PCa and transition to CRPC [22]. Ongoing studies are attempting to identify these pathways that promote androgen signaling especially in low/no androgen backgrounds and the receptors that trigger these pathways.

In recent years, three putative membrane androgen receptors have been proposed, viz. GPRC6A [23], OXER1 [24] and ZIP9 [25], of which the first two are GPCRs. While GPRC6A mediates most of its functions by activating the Gq and Gi signaling pathways [26], OXER1 functions are mediated via $G_i$ or $G_{\beta/\gamma}$ [24] pathways. Neither of these GPCRs are known to play a role in classical androgen signaling.

Here we identified an adhesion GPCR, GPCR 205 (GPR56) with structural similarity to the AR ligand binding domain (LBD) using a bioinformatics approach. Further, we analyzed it as a candidate for membrane androgen binding. Our results demonstrate that androgen-induced activation of GPR56, is required for complete activation of the AR. Compared to normal tissue GPR56 mRNA expression was higher in all 25 tumor samples tested. Our results provide an alternate mechanism of AR activation in PCa patients, via GPR56 and this may be exploited to aid the hormonal therapies in CRPC patients.

## Materials and methods

### Cell culture

LNCaP, PC3, DU145, TM3, HEK293 and Cos-1 cell lines originating from ATCC were purchased from National Centre for Cell Science (NCCS) (Pune, India). These cells were

maintained in RPMI 1640, DMEM and HAM'S F12 (Himedia, New Delhi, India) with 10% Fetal bovine serum (Thermo Fischer Scientific, USA). Trypsin EDTA, other molecular biology grade chemicals were obtained from SRL, India and Lipofectamine 2000 was obtained from Thermo Fischer Scientific (USA).

## Computational modelling and *insilico* docking studies

The identification of candidate proteins which can bind with high affinity to testosterone was carried out using a computational approach which is based upon the three-dimensional structure models of candidate proteins. The structures of selected GPCRs were modelled using I-TASSER software at server (https://zhanglab.ccmb.med.umich.edu/I TASSER/). All the generated models were subjected to PROCHECK to select the best model using the statistics of Ramachandran Plot. The models which displayed cut off greater than 97% of residues in favored regions and less than 3% residues in disallowed region were selected. The best models generated for candidate GPCRs were submitted to Protein Model Data Base (PMDB) at server (http://bioinformatics.cineca.it) and the PMDB IDs are (GPCR 205-PM0081560, GPCR 47-PM008561, GPCR 231-PM0081562, GPCR 232-PMOO81563, GPRC6A- PM0081564). The interaction studies of modelled GPCR with ligand testosterone was performed using Schrodinger Glide Maestro, USA software.

## Constructs

Human GPR56 NT (amino acids 1–510) and GPR56 CT (amino acids 511–693) were cloned into pcDNA3.1 at the Hind III site and the primers used for N terminal were 5'AGTGAAG CTTATGCCTCAGCCTCC-3' and 5'-GTGCAGAAGCTTGGCACTGGCTCG-3' respectively, while those of GPR56 CT were 5'-TGCAAAGCTTCCTGCTCACCTGCC-3' and 5'-GGTAA GCTTGAGGCCTAGATGCG-3' respectively. I626 and W623 double mutant receptor was generated using the following primers 5'- GCCTGCCCTGGGCCTTGATCTTCTTCTCCT-3' and 5' GACCAGGCTGAGGCCCAGCAGTGTCAGCACA-3'. The mutant was generated using the Q5 ® Site-Directed Mutagenesis kit (NEB, catalog number #E0554S) according to the manufacturer's protocol.

## RNA isolation and reverse transcription PCR

Total cell RNA was isolated for GPR56 gene expression analysis using Trizol based method as per manufacturer's protocol. Cells were lysed on petri dishes using 1 ml of Trizol Reagent. Total tissue RNA was isolated from normal and tumor patient's tissue using Trizol Reagent. 1 ml of Trizol reagent was added to 50–100 mg of tissue and was sonicated for 25 min. Chloroform was added followed by vigorous shaking for 15–30 sec and left at RT for 5 min. The solution was then centrifuged at 12000 g for 15 min at 4˚C. The top aqueous layer was carefully removed and precipitated by adding 500 µl of isopropanol for 10 min at RT. The precipitated RNA was then pelleted by centrifugation at 12,000 rpm for 10 min at 4˚C. The pellet was air dried and resuspended in 50µl RNase free DEPC water. 1 µg of RNA obtained was used for cDNA synthesis using Biorad Iscript cDNA synthesis kit (as per manufacturer's protocol) and used subsequently for quantitative RT-PCR analysis for GPR56 expression. For amplification of GPR56, PCR reaction was performed using the following primer pair 5' GGTACAGAACA CCAAAGTAGCCAAC (Forward) 3'-TCAACCCAGAACACACATTGC (Reverse) (synthesized by Gbiosciences, USA). For amplification of GPR56 gene (N-terminal and C-terminal) in normal and tumor tissue following primer pairs were used (synthesized by eurofins Genomics, Karnataka, India): GPR56 (N-Terminal), 5'-GAGTAGCTGGGATTACAGGT (Forward) and 3'-AAGCGAAAGTCTTCCCTGTG (Reverse) and GPR56(C-Terminal), 5'-CTCGAGGGGTAC

AACCTCTA (Forward) and 3' – TAGTTGTCCACATCCACCAGG (Reverse). Beta-actin expression was used as Control. Reaction products were resolved on 2% agarose gel to determine the molecular sizes of the GPR56 amplicons. The gel images were obtained using BIO-RAD molecular imager.

## Western blot analysis

Whole cell lysate was prepared using cell lysis buffer containing 20mM Tris/HCl (pH 7.5), 150mM NaCl, 1% Nonidet P40, 1mM Dithiothreitol with protease inhibitors aprotinin (10μg), Phenyl methane Sulfonyl fluoride (10μg) and phosphatase inhibitor cocktail (10μl). The lysate was centrifuged at 14,000rpm at 4°C for 15 mins. The pellet was discarded, and supernatant was used as whole cell lysate. The protein concentrations were measured using Bradford kit (Himedia). The samples were separated by SDS/PAGE and transferred to PVDF membrane (MDI). The membrane was blocked in TBST (Tris buffered saline/Tween 20: 20mM Tris/HCl ph-7.5, 150mM NaCl, 0.1% Tween 20) with 5% Non-fat dry milk for 2 hours at 4°C. The membrane was incubated with primary antibody diluted in TBST with 3% non-fat dry milk overnight at 4°C. Membrane was washed thrice with TBST for 15 mins each and incubated with HRP-conjugated secondary antibody for 2 hours at RT. Post washing, immunoreactive bands were detected by femtoLUCENT (G Biosciences). Primary AR antibody (AR 441), Primary GPR56 antibody and HRP conjugate secondary antibody were purchased from Santa Cruz Biotechnology Ltd, Histone H3B antibody was a kind gift provided by Dr. Sandeep Saxena from National Institute of Immunology, India.

## Flow cytometry

LNCaP or HEK 293 cells transfected with GPR56, siRNA GPR56 and scrambled RNA (GPR56) were detached by scraping, washed once with PBS containing BSA and suspended in phosphate-buffered saline containing BSA. Live cells were identified by trypan blue staining. The cells were then incubated for 10 min with BSA FITC (1M) or T-BSA FITC (1μM) for detecting membrane androgen binding sites and non-specific binding. Cells were analyzed by flow cytometry using BD LSR II Flow cytometer and using BD FACS Diva v 8.0.1 software in a sample size of 10,000 cells gated.

## siRNA transfection

LNCaP and HEK293 were cultured and transfected with siGPR56 (100nM)/siPKA(120nM) using Lipofectamine 2000 (Thermo Fischer Scientific, USA) in serum free media. Fresh serum containing media was added 24 hrs later. Treatments were done after 24 hrs of media change.

## GTP-Rho pulldown assay

HEK293 cells were transfected with GPR56, Gα13 siRNA, scrambled siRNA, GPR56 [C], GPR56[N], Cells at 30% confluency were serum starved for 24hrs and treated with testosterone for 24 hrs at 10nM and H89 for 30 mins at 30μM, GTPγS was used as positive control. The cells were then lysed in 300μl of cell lysis buffer (provided with kit, 50 mM Tris p H 7.5, 10 mM $MgCl_2$ 0.5 M NaCl and 2% Igepal) per 90 mm dish. After centrifugation at 15,000 x $g$ at 4°C for 10 min, 200 μl of supernatant was incubated with 30μg of rhotekin (GST-RBD) beads at 4°C for 1 hour on rocker. The resins are then washed twice with 500 μl of wash buffer (25 m M Tris p H 7.5, 30 mM $MgCl_2$, 40 mM NaCl and boiled for 2 min in Laemmli's sample buffer and bands were detected using anti-RhoA mAb (cytoskeleton ARH03, 1:500) by femtoLUCENT (G Biosciences).

## PKA activation assay

To check the androgen signaling and GPCR mediated PKA activation, HEK 293 cells were transfected with clones of GPCR 205 (human) and GPRC6A (human). The GPCR 205 (human) clone was obtained from Dr. Hsi-Hsein laboratory, Chang Gung University, Taiwan. Human GPRC6A clone was obtained from Prof. Hans Brauner Laboratory, University of Copenhagen, Denmark. The cells were changed to serum-starving medium 24 hrs after transfection followed by T treatments at final concentration of 100nM at different time points (20 mins and 1 hr) also positive control (isoproterenol) treatment (10min) at final concentration of 10 µM. The PKA ELISA kit uses specific antibody for PKA and detect the PKA activation through chemiluminescence. These cells were then washed with TBS buffer and fixed with fixing solution. All other steps were carried out according to the manufacturer instructions. Chemiluminescence were detected by taking OD at 450 nm using ELISA Plate Reader. The $OD_{450}$ values were normalized with the average values obtained for GAPDH and the data was shown as the relative PKA activity compared to control.

## Cyclic AMP assay

Cells were seeded equally (5000 cells per well) in 96-well plates and allowed to grow overnight. The HEK 293 cells and LNCaP cells were serum starved for 24 hours followed by stimulation with testosterone (10nM and 100nM) for 10 min at 37˚C. Isoproterenol (10 µM) was used as positive control in both cell lines. The cAMP assay was performed using the cAMP-Glo™ Assay Kit (Promega) according to the manufacturer's instructions. The assay was performed in triplicate and repeated three times.

## Live cell imaging

LNCaP cells cultured in 100mm petridishes with cover slips and then transfected with GFP-AR plasmid on sterile glass with and without siGPR56 (siRNA against GPR56). The next day, cells were exposed to 10nM testosterone. Hoechst dye was used for nuclei staining. Fluorescence imaging of live cells was performed through an upright Olympus Optical Microscope. At different time intervals of 5 mins, 15 mins, 30 mins, 1 hour, 4 hours and 24 hours cells were visualized under microscope by mounting the coverslip on microslide.

## Luciferase reporter assay

LNCaP and HEK293 cells were cultured in media with 10% FBS and transfected using Lipofectamine 2000. Transfections were carried out with PSA-Luc plasmid (1µg). After 24 hours of transfection the cells were equally splitted in to 60mm petridishes and allowed to attach. After 24 hours, the identical cell population were treated with testosterone (0.1 and 10nM). Luciferase activities were measured with Luciferase assay kit (Promega) and were normalized with protein concentrations of samples.

## Tissue specimens

Twenty-five prostate tumor and matched normal tissues were obtained from radical prostatectomy specimens (S1 Table). This study was reviewed and approved by the Institutional Review Board (IRB) of Rajiv Gandhi Cancer Institute and Research Centre, New Delhi, India (586/SO/SKR-49). All methods were performed in accordance with the relevant guidelines and regulations. Matched normal tissues were obtained from the lateral lobe of the same specimen. Matched normal (N) and tumor tissue were stored in RNA Later at $-80^{0}$ C immediately after surgical excision. All patient's information was anonymized prior to analysis.

## MTT assay

The MTT (3-(4,5-dimethylthiazol-2-yl)-2,5-diphenyl tetrazolium) assay was used to assess cell proliferation using the Cell Titer 96® AQ$_{ueous}$ One Solution Cell Proliferation kit, Promega, USA. Cells were seeded in 96-well plates (5000 cells/well) for 1, 2, 3, and 4 days and 20 μL of assay reagent was added into the medium (as per the manufacturer's instructions) and incubated for 4 hours at 37˚C in humidified 5% $CO_2$ incubator. The absorbance at 490 nm was measured on each day using ELISA plate reader (Lisa Plate Reader). Each experiment was repeated for at least three times.

## Statistical analysis

Experiments were repeated atleast three times. All statistical analysis was performed using Prism (Version 5.0) as detailed in the results. ($p < 0.05$ was retained as a significance threshold). Data are expressed as mean ± SD. All statistical comparisons were made by using one-way ANOVA and two-way ANOVA test.

# Results

## 1.Sequence comparison of mammalian GPCRs with the Ligand Binding Domain (LBD) of AR

BLAST analysis was performed using the cDNA sequence of all (826) mammalian GPCRs and the sequence of Ligand Binding Domain (LBD) of AR. The candidates which demonstrated highest percentage identity (91–100%) and query cover (26–39%) were shortlisted namely GPCR 205, GPCR 231, GPCR 232, GPCR 47, GPCR 708 and GPCR 675 (Table 1).

The identity and query cover of GPCRs 215 and 663 proposed as putative membrane androgen receptor earlier have also been included. GPCR 231, Leucine-rich repeat-containing G-protein coupled receptor 4 (G- protein coupled receptor 48); GPCR 205, G-protein coupled receptor 56 (Protein TM7XN1); GPCR 47, gamma-amino butyric acid (GABA) B receptor subunit 1, GPCR 232, Leucine-rich repeat-containing G-protein coupled receptor 5 (G- protein coupled receptor 49) (G-protein coupled receptor 67) (G- protein coupled receptor HG38); GPCR 663, Oxoeicosanoid receptor 1 (5-oxo-ETE G-protein coupled receptor) (G-protein coupled receptor 170) (G-protein coupled receptor R527) (G-protein coupled receptor TG1019); GPCR 708, Probable G-protein coupled receptor 179 (Probable G-protein coupled receptor 158-like 1) (GPR158-like), GPCR 675, Probable G-protein coupled receptor 110 (G-protein coupled receptor KPG_012) (G-protein coupled receptor PGR19), GPCR 215,

**Table 1. Mammalian GPCRs that exhibit greater than 91% identity and 25% query cover compared to the LBD of AR were shortlisted.**

| S. NO. | ACCESSION NO. | SEQUENCE TESTED | %IDENTITY | QUERY COVER | DETAILS |
|---|---|---|---|---|---|
| 1. | Q9BXB1 | NUCLEOTIDE | 100% | 28% | GPCR 231 |
| 2. | Q9Y653 | NUCLEOTIDE | 100% | 27% | GPCR 205(GPCR 56) |
| 3. | Q9UBS5 | NUCLEOTIDE | 100% | 26% | GPCR 47(GABA B RECEPTOR SUBUNIT 1) |
| 4. | O75473 | NUCLEOTIDE | 95% | 29% | GPCR 232 (GPCR 49) |
| 5. | Q6PRD1 | NUCLEOTIDE | 91% | 39% | GPCR 708(Probable GPCR 179 Like) (Probable GPCR 158 Like) |
| 6. | Q5T601 | NUCLEOTIDE | 91% | 29% | GPCR 675(Probable GPCR 110) (GPCR PGR19) ORPHAN RECEPTOR |
| 7. | Q5T6X5 | NUCLEOTIDE | 90% | 18% | GPCR 215 (G-protein coupled receptor family C group 6 member A) (hGPRC6A) |
| 8. | Q8TDS5 | NUCLEOTIDE | 94% | 6% | GPCR 663 (Oxoeicosanoid receptor 1) |

G-protein coupled receptor family C group 6 member A (hGPRC6A) (G-protein coupled receptor GPCR33) (hGPCR33), mAR, Membrane Androgen Receptor.

The identity and query cover of two putative androgen responsive GPCRs identified earlier, namely, GPRC6A [23] and OXER1 (GPCR 663) [24] were also determined (Table 1). Models for selected GPCRs were designed using I-Tasser software (Fig 1A), in absence of crystal structures and best models were selected through Ramachandran Plot (Fig 1B). GPCR 708 (containing 2367 amino acids) could not be modelled as its size was beyond the acceptable range of the I-Tasser software. Molecular docking using models of target proteins with testosterone (T) was carried out using the Schrodinger Glide Maestro, USA software (Fig 1C). GPCR 205 (GPR56) [27] exhibited the highest docking score of -8.3 Kcal (Table 2) and was used for further studies.

The critical residues, capable of interacting with testosterone owing to their strategic location in the binding pocket include I626, W623, W563 and N574 and are displayed in residue interaction diagram (Fig 1D). The residues I626 and W623 were capable of covalent

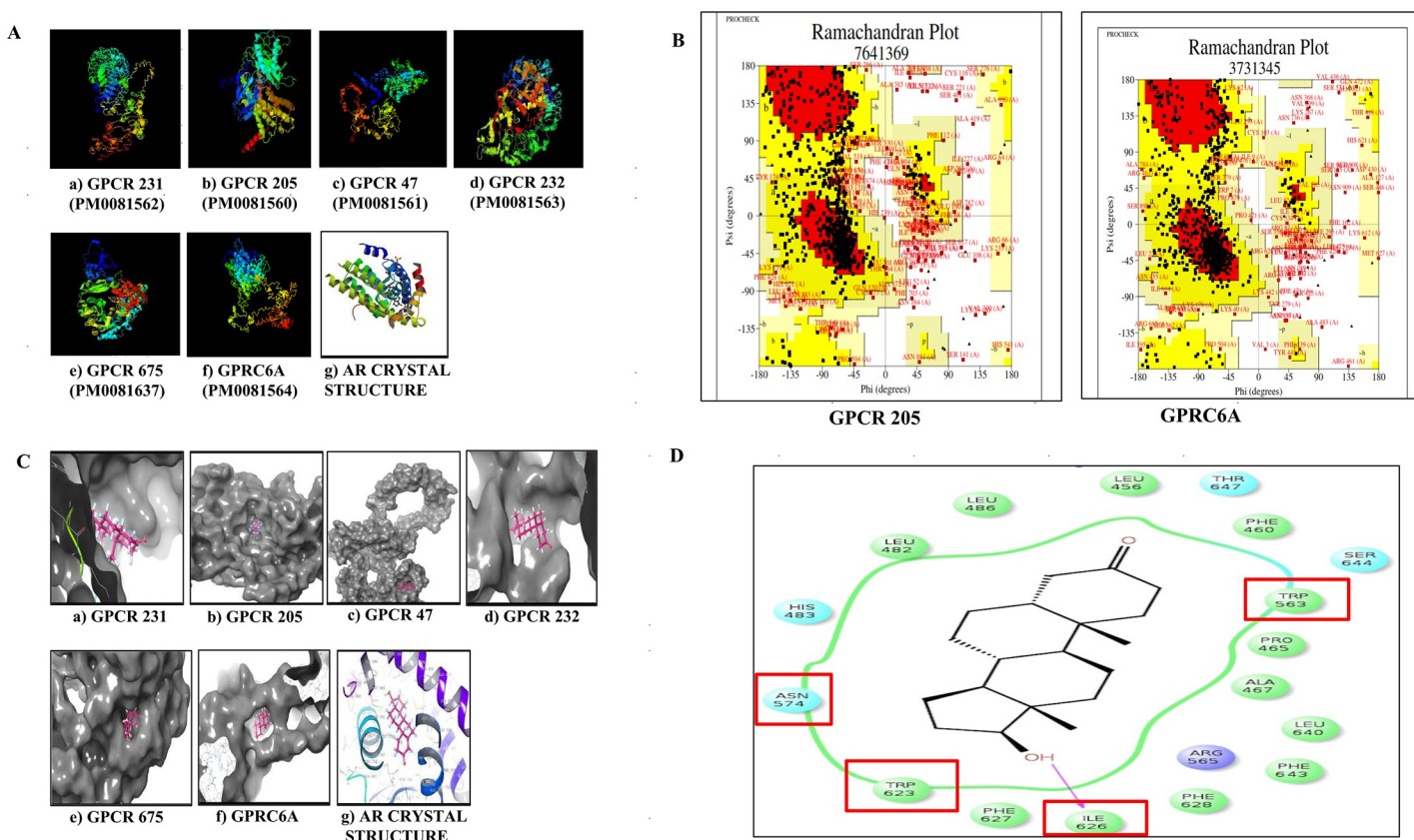

**Fig 1. Sequence comparison of mammalian GPCRs and the ligand binding domain of AR.** A) The 3D structures of candidate GPCRs as predicted using I-TASSER. The 3D models of the target protein were deposited to the Protein Model Database (PMDB) and each model was assigned a unique PMDB model identifier ID. B) Example of Ramachandran plots for selection of best model of the GPCRs (i.e. for GPRC6A and GPCR 205 (Ramachandran plot for other GPCRs not shown). The models for the GPCRs were selected based on the maximum residues obtained in the most favored region (MFR) using PROCHECK software. C) *In silico* molecular docking of selected GPCRs with testosterone (ligand) using Schrodinger Glide Maestro. Using multiple sequence alignment (clustal W) the conserved regions in selected GPCRs were identified. Testosterone was docked at the conserved region of the target GPCR to have a specific region docking prediction. D) Residue interaction diagram using Schrodinger Maestro software showing residues surrounding testosterone. The critical residues capable of making covalent interactions with testosterone are I626, W623, W563, N574. The residues I626 and W623 were mutated into Proline and Arginine using site directed mutagenesis for mutation experiments.

**Table 2. Docking scores and energy minimization of target proteins.** Molecular docking using models of receptor (protein) and ligand (Testosterone) was carried out using Schrodinger Glide Maestro, USA software. GPCR 205 displayed highest docking score of -8.3Kcal/mol.

| S.NO. | POTENT TARGET | XP DOCKING SCORE(Kcal/mol) | ENERGY MINIMIZATION |
|---|---|---|---|
| 1. | Androgen Receptor (crystal structure) | - 12.336 | -55311.2461 KJ/mol |
| 2. | GPCR 205 (GPR56) | - 8.324 | -123777.8203 KJ/mol |
| 3. | GPRC6A | - 5.886 | -1,33,107 KJ/mol |
| 4. | GPCR 232 | - 5.6 | -147342.0156 KJ/mol |
| 5. | GPCR 231 | - 5.3 | -1,61716.3438KJ/mol |
| 6. | GPCR 47 | - 5.5 | -59662.6875KJ/mol |
| 7. | GPCR 663 | -5.011 | -1,61885.37KJ/mol |
| 8. | GPCR 675 | -4.460 | -165970.6562KJ/mol |

interactions with testosterone and were therefore mutated into proline and arginine respectively, using site directed mutagenesis for verifying the interactions of testosterone with GPR56. A phylogenetic tree was also created to check the relatedness of the different GPCRs and AR (S1 Fig).

GPR56 belongs to the adhesion G-protein coupled receptor family [28] and mutations in GPR56 result in a severe brain malformation known as bilateral frontoparietal polymicrogyria (BFPP) [27]. The known ligands of GPR56 in cultured cells includes collagen III, TG2 and heparin which activate downstream Rho signaling [29].

## 2. GPR56 expression in cell lines and membrane androgen binding

The adhesion GPCR family members are expressed in almost all-important organ systems of the body and play a role in neuronal functions, immunity, reproduction and development [30]. The expression of GPR56 at mRNA and protein levels was determined in three prostate cancer cell lines, namely, LNCaP, PC3, DU145 and two non-prostate cell lines, TM3 and HEK293 were chosen. RT-PCR analysis with RNA from these cell lines revealed that GPR56 was expressed in all these cell lines except HEK293 (Fig 2A). Corresponding to this, GPR56 protein expression was observed in all prostate cell lines and the non-prostate cell line TM3 (Fig 2B). Robust GPR56 protein expression was observed in LNCaP cells, which was reduced significantly upon transfection with GPR56 siRNA (Fig 2C).

To confirm the binding of testosterone to GPR56, flow cytometry was performed in LNCaP cells that express GPR56 mRNA. Correlating with the observed mRNA expression, LNCaP cells exhibited 88.3% binding to T/BSA/FITC (Fig 2D). This binding was reduced by 69.2% upon transfection of the LNCaP cells with GPR56-siRNA (Fig 2E). There was no reduction in binding upon transfection of LNCaP cells with scrambled siRNA (Fig 2F).

In (human) GPR56 transfected -HEK293 cells, incubation with T/BSA/FITC for 10 minutes caused 93.5% binding whereas it was 13.6% in non-transfected cells (Fig 2G and 2H) correlating with negligible GPR56 mRNA expression observed in these cells. In HEK293 cells transfected with both GPR56 and GPR56 siRNA, T/BSA/FITC binding was 19.1% while it was 94% in cells transfected with GPR56 and scrambled siRNA (Fig 2I and 2J).

## 3. GPR56 mutants and FACS

To confirm that testosterone was indeed binding to the C-terminal domain of GPR56, as predicted by the bioinformatics analysis, GPR56 N-terminal (1–509), C-terminal (511–693) mutants and the double mutant (where Ile 626 & Trp 623 had been mutated) were also transfected into HEK 293 cells and FACS analysis was performed (Fig 3A). Interestingly, binding of

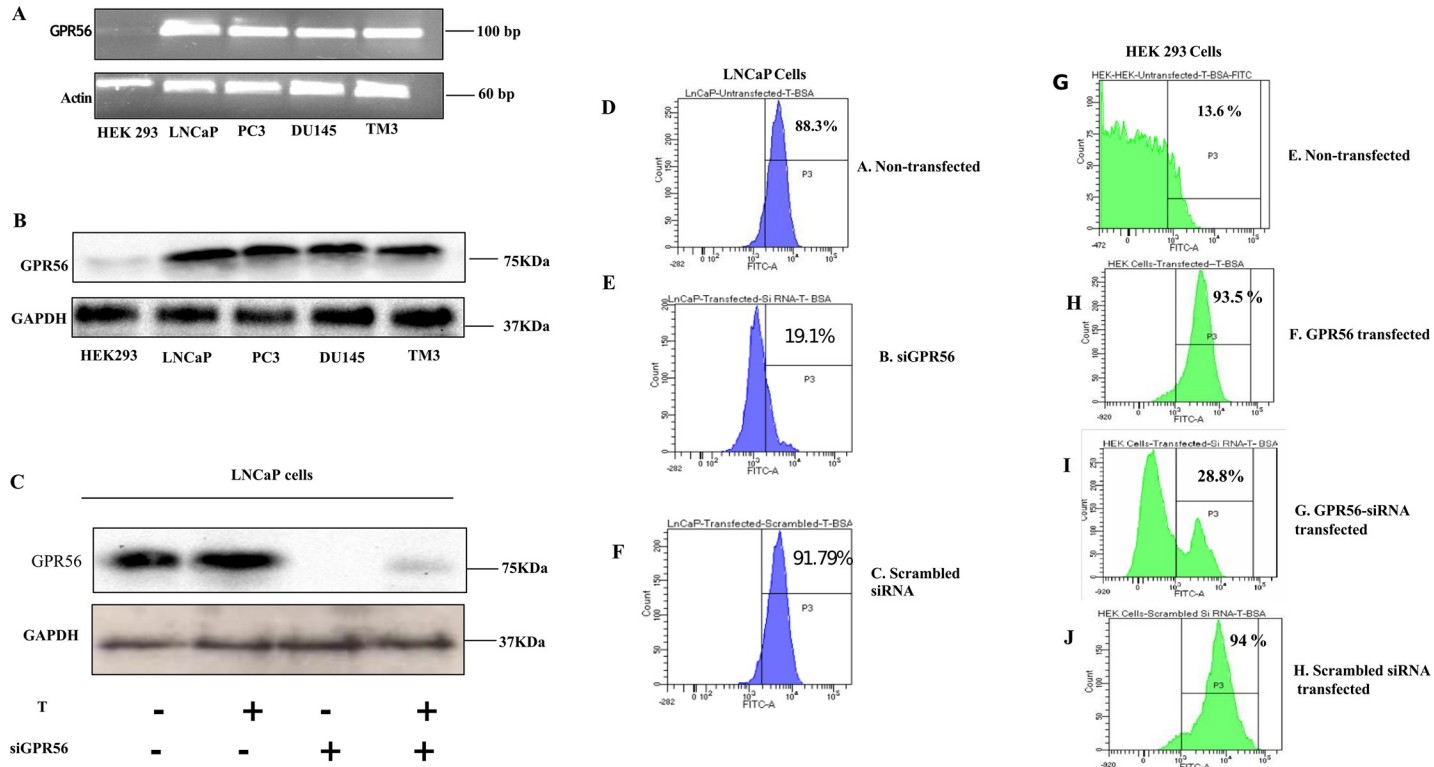

**Fig 2. GPR56 expression in cell lines and membrane androgen binding.** A) RT-PCR was performed using total RNA extracted from LNCaP, PC3, DU145, HEK293 and TM3 cell lines using internal primers for GPR56. Beta actin was used as a control B) GPR56 protein expression detection in prostate and non-prostate cell lines (LNCaP, PC3, DU145, HEK293 and TM3) through immunoblotting. Western blotting was done using 50 ug of protein from cells lysate using GPR56 antibody. Glyceraldehyde-3-phosphate dehydrogenase (GAPDH) was used as a loading control C) Reduction of GPR56 protein expression in LNCaP cells transfected with GPR56-siRNA was detected using immunoblotting. GAPDH was used as a loading control. D-J) Flow cytometry analysis for detection of membrane androgen binding sites in LNCaP and HEK293 cell lines. Cells were labelled with T-BSA-FITC and the percentage of cells that bind specifically are given by Gate P3 and % parent shown below the figure. Membrane testosterone binding sites on LNCaP cells and cells transfected with GPR56-siRNA or scrambled siRNA were analyzed. Flow cytometry analysis was also performed in GPR56 transfected and non-transfected HEK 293 cells, and GPR56 overexpressing HEK293 cells transfected with GPR56-siRNA or scrambled siRNA. Cells were labelled with BSA-FITC as a control. Figures are representative of three independent experiments.

T-BSA-FITC to non-transfected HEK293 cells was negligible (Fig 3B), whereas binding to GPR56 was 99. 4% (Fig 3C). The N-terminal mutant and the GPR56 double mutants also exhibited negligible binding (Fig 3D and 3E), however, the GPR56 C-terminal exhibited 98.6% binding, comparable with the binding of GPR56 wildtype (Fig 3F). The FACS results confirmed that testosterone bound to the C-terminal domain of GPR56 as predicted. Multiple Sequence Alignment was performed to assess the conservation of Ile 626 and Trp 623 in GPR56 from different species and also among adhesion GPCRs of family VIII to which GPR56 belongs (Figs 4 and 5).

The Ile 626 was replaced by another nonpolar amino acid residue Valine in other species (or Glycine in one species) whereas the Trp 623 remained unchanged in all species studied.

## 4. GPR56 regulates AR transactivation

To investigate whether GPR56 activation could contribute to classical androgen receptor signaling, ARE-regulated transcription of luciferase reporter gene was measured in GPR56 expressing LNCaP cells. In these cells, treatment with 0.1nM and 10nM Testosterone stimulated AR activity 2.6 and 4 folds respectively. However, negligible activation was seen in LNCaP cells transfected with siRNA against GPR56 or PKA (Fig 3G). AR activity fold changes

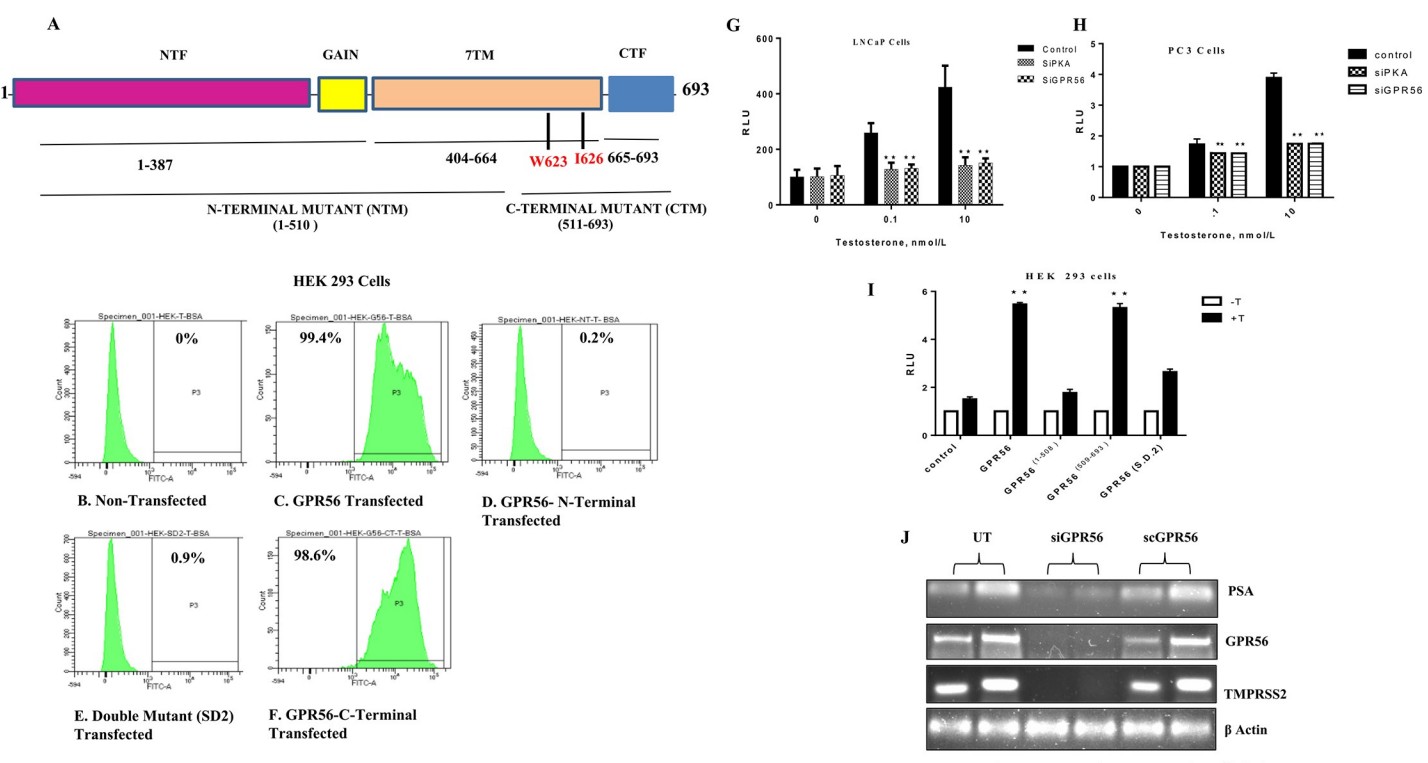

**Fig 3. Binding of GPR56 mutants to testosterone and AR transactivation.** A) Representative picture of 1–693 aa sequence of GPR56 showing important domains and sequences included in mutants. The N-terminal fragment (NTF) is linked to C-terminal fragment (CTF) through GAIN domain and 7 transmembrane domain B-F) Flow cytometry analysis of HEK293 cells transfected with wild type GPR56, GPR56 NT mutant, GPR56 CT mutant and GPR56 double mutant and treated with T-BSA-FITC to assess binding. Cells were labelled with T-BSA-FITC and the percentage of cells that binds specifically are given by gate P3 and % parent are also shown. Cells were labelled with BSA-FITC as a control. Figures are representative of three independent experiments. G) Inhibition of AR transcription by knocking down GPR56 and PKA expression. LNCaP cells transfected with ARE-Luc (1μg) reporter plasmid with scrambled siRNA(100nM) or siGPR56 (100nM) and siPKA (120nM). The cells were treated with 0.1 and 10nM testosterone 24 h after transfection period. Values are mean ±S.D. from three independent experiments. * p< 0.01, ** p<0.001, two-way anova test. H) PC3 cells transfected with ARE-Luc (1μg) reporter plasmid with scrambled siRNA (100nM) or siGPR56 (100nM) and siPKA (120nM). The cells were treated with 0.1 and 10nM testosterone 24 h after transfection period. Values are mean ±S.D. from three independent experiments. * p< 0.01, ** p<0.001, two-way anova test. I) AR transcription analysis of GPR56-NT, GPR56-CT, and GPR56 double mutant. HEK293 cells transfected with ARE-Luc (1μg) reporter plasmid with GPR56, GPR56 N terminus mutant, GPR56 C terminus mutant, GPR56 double mutant (SD2) plasmids. The cells were treated with 10nM testosterone 24 h after transfection period. Values are mean ±S.D. from three independent experiments. * p< 0.01, ** p<0.001, two-way anova test. J) GPR56 regulates expression of PSA & TMPRSS2. LNCaP cells transfected with scrambled siRNA (Sc GPR56) or siRNA against GPR56 (siGPR56), were treated or not treated with testosterone (10nM)for 24 hrs. Total RNA was isolated and expression of PSA & TMPRSS2, AR target genes was measured by RT-PCR, beta-actin was used as control.

were same as in wildtype when scrambled siRNA was used. Similarly, in PC3 cells on treatment with 0.1 n M and 10 nM Testosterone increased AR activity 1.9 and 4-fold respectively. However, negligible activation was seen in PC3 cells transfected with siRNA against GPR56 or PKA (Fig 3H). Consistent with this, HEK293 cells overexpressing GPR56 exhibited five-fold increase in luciferase activity when treated with 10 nM T, compared to non-transfected cells (Fig 3G). Also, the luciferase activity of the HEK293 cells transfected with GPR56 N-terminal, C-terminal and double mutant were 1.7 folds, 5.3 folds and 2.6 folds respectively as compared to that transfected with vector alone (Fig 3I).

Further, we evaluated the effect of GPR56 knockdown on the transcription of PSA and TMPRSS2 mRNA, as prototype AR target genes in LNCaP cells. RT-PCR analysis demonstrated that expression of both PSA and TMPRSS2 was enhanced significantly in LNCaP cells or scrambled siRNA-transfected LNCaP cells upon testosterone treatment but was reduced significantly in LNCaP cells transfected with GPR56 siRNA (Fig 3J). Interestingly, the

| Species | Multiple Sequence Alignment (Amino Acids) | |
|---|---|---|
| Poeciliopsis | ---------------------------------------------------------- | 179 |
| Danio | LMGFGFPFLLVSILLSVG-DIYGERKIK--PSDDVNNPYRMC**W**MTEGDKSQLAHYII**N**IG | 535 |
| Cricetulus | IVGWGFPVFLVTLVALVDVNNYGPIILAVRRTPDRVIYPSMC**W**IRDS----LVSYVT**N**LG | 570 |
| Rattus | TVGWGFPVFLVTLVALVDVNNYGPIILAVRRTPDHVIYPSMC**W**IRDS----VVSYVT**N**LG | 570 |
| Homo | AMGWGFPIFLVTLVALVDVDNYGPIILAVHRTPEGVIYPSMC**W**IRDS----LVSYIT**N**LG | 576 |
| Bos | IVGWGFPASLVMLVALVDVNNYGRIILAVHKTPESVIYPSMC**W**IQDS----LVSHVT**N**LG | 569 |

| Species | Multiple Sequence Alignment (Amino Acids) | |
|---|---|---|
| Poeciliopsis | ---------------------------------------------------------- | 179 |
| Danio | LLAVVVSSGLVMLFLVV-REIRNRPDWKKIHVAFLSIWGLTCLYGTT**W**AL**G**FLDF—GPF | 592 |
| Cricetulus | LFSLVFLFNMAMLATMVVQILRLRPHNQK-WPHVLTLLGLSLVLGLP**W**AL**V**FFSFASGTF | 629 |
| Rattus | LFSLVILFNMAMLATMVVQILRLRPHSQK-WPHVLTLLGLSLVLGLP**W**AL**V**FFSFASGTF | 629 |
| Homo | LFSLVFLFNMAMLATMVVQILRLRPHTQK-WSHVLTLLGLSLVLGLP**W**AL**I**FFSFASGTF | 635 |
| Bos | LFSLVFLFNTAMLGTMVVQILRLRPHAQK-WPHVLTLLGLSLVLGLP**W**AL**V**FFSFASGTF | 628 |

**Fig 4. Multiple Sequence Alignment (MSA) of full length GPR56 in six different species.** I626 (highlighted in yellow), W623 (highlighted in pink), N574 (highlighted in green), and W563 (highlighted in blue) of GPR56 are involved in making good interaction with Testosterone are highly conserved in 5 different species.

| Adhesion GPCR | Multiple Sequence Alignment (Amino Acids) | |
|---|---|---|
| GPR112 | HDLKGTMSLTFLLGLT**W**GF**A**FFAW--GPMRNFFLYLFAIFNTLQGFFIFVFHCVMKESVR | 2994 |
| GPR126 | RNLRSVVSLTFLLGMT**W**GF**A**FFAW--GPLNIPFMYLFSIFNSLQGLFIFIFHCAMKENVQ | 1122 |
| GPR64 | QDLRSIAGLTFLLGIT**W**GF**A**FFAW--GPVNVTFMYLFAIFNTLQGFFIFIFYCVAKENVR | 887 |
| GPR97 | KKVLTLLGLSSLVGVT**W**GL**A**IFTP----LGLSTVYIFALFNSLQGVFICCWFTILYLPSQ | 529 |
| GPR56 | SHVLTLLGLSLVLGLP**W**AL**I**FFSFASGTFQLVVLYLFSIITSFQGFLIFIWYWSMRLQAR | 666 |
| GPR114 | HDTVTVLGLTVLLGTT**W**AL**A**FFSF--GVFLLPQLFLFTILNSLYGFFLFLWFCSQRCRSE | 511 |

| Adhesion GPCR | Multiple Sequence Alignment (Amino Acids) | |
|---|---|---|
| GPR112 | PTTPFC**W**IKDDSI---FYIS**V**VAYFCLIFLMNLSMFCTVLVQLNSVKSQI-QKTRRKMIL | 2936 |
| GPR126 | KGDEFC**W**IQDPVI---FYVT**C**AGYFGVMFFLNIAMFIVVMVQICGRNGKRSNRTLREEVL | 1064 |
| GPR64 | SPDDFC**W**INNNAV---FYIT**V**VGYFCVIFLLNVSMFIVVLVQLCRIKKKKQLGAQRKTSI | 829 |
| GPR97 | TSLELC**W**FREGTTMYALYIT**V**HGYFLITFLFGMVVLALVVWKIFTLSRATAVK-ERGKNR | 473 |
| GPR56 | IYPSMC**W**IRDSLV---SYIT**N**LGLFSLVFLFNMAMLATMVVQILRLR-------PHTQKW | 606 |
| GPR114 | QNMSIC**W**VRSPVV---HSVL**V**MGYGGLTSLFNLVVLAWALWTLRRLRERADA--PSVRAC | 453 |

**Fig 5. Multiple Sequence Alignment of full-length adhesion GPCRs-Group VIII through clustal omega.** I626 (highlighted in green), W623 (highlighted in red), N574 (highlighted in pink) and W563 (highlighted in yellow) of GPR56 involved in making good interaction with testosterone are highly conserved in all adhesion GPCRs of group VIII.

expression of GPR56 was increased in Testosterone-treated LNCaP cells or cells transfected using scrambled siRNA.

## 5. Androgens activate PKA via GPR56

As GPR56 contributed to increased AR downstream signaling in prostate cells, it was apparent that androgen binding to GPR56 activated a downstream signaling pathway that contributed to classical androgen signaling. Since activation of PKA is a prerequisite step for AR transcription [31], we investigated whether testosterone treatment of GPR56 containing cell enhanced PKA. Aberrant activation of PKA is observed in many CRPC patients [21]. Recently it has been demonstrated that activation of PKA is necessary for release of AR from HSP90 in the cytoplasm and its subsequent migration into the nucleus [31]. Also, in LNCaP cells that express GPR56, testosterone induced PKA activation was greater than three folds as compared to control, whereas in LNCaP cells transfected using GPR56-siRNA or PKA-siRNA, testosterone induced PKA activation was ~ 1.5 and 1.3 folds respectively (Fig 6A). Fold changes in PKA activity were same in wildtype when scrambled siRNA was used. In HEK 293 cells that

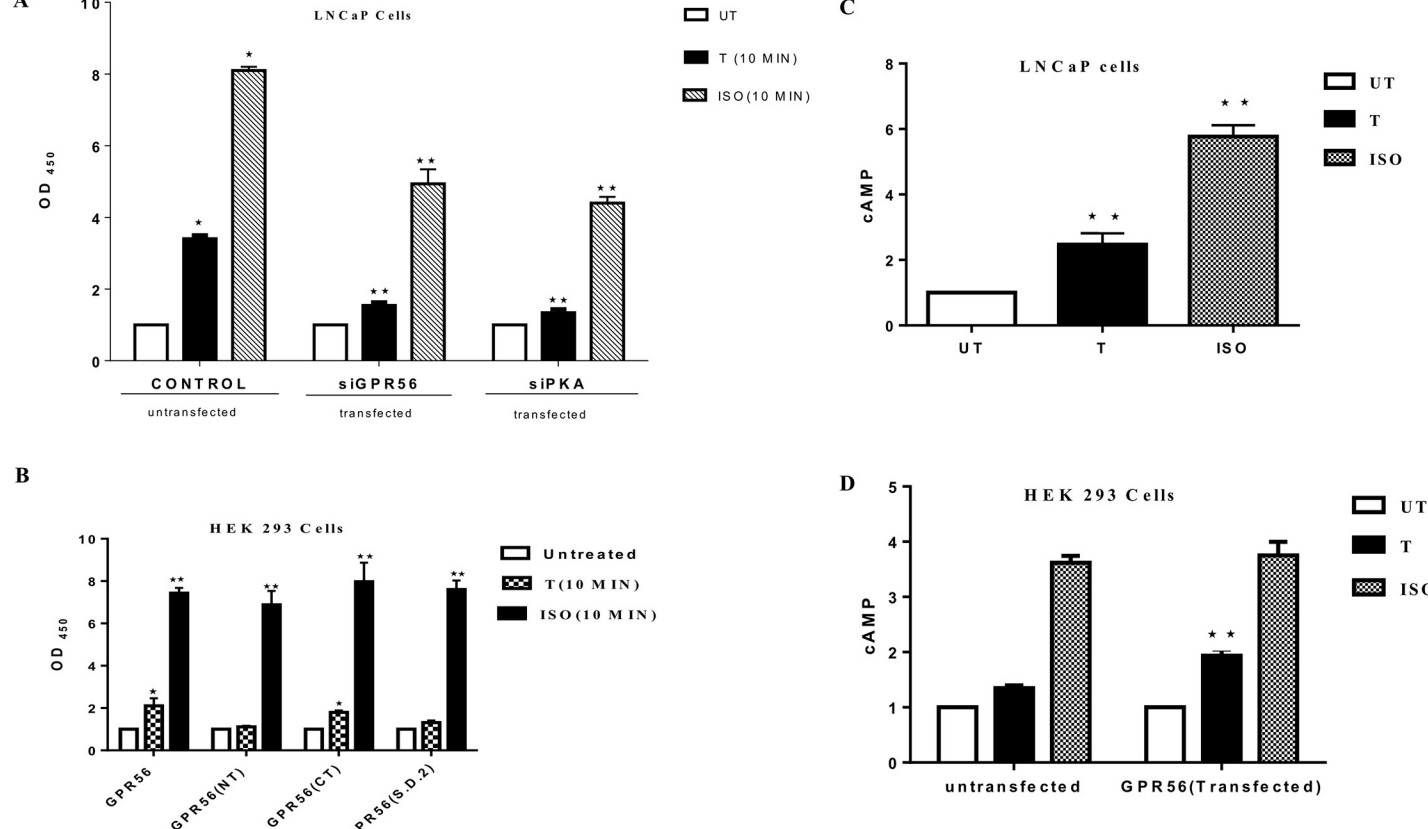

**Fig 6. Androgen-stimulation of GPR56 activates the PKA pathway.** A) PKA activity in LNCaP cells. PKA activity of GPR56-siRNA or siPKA transfected LNCaP cells was determined and compared to that of untransfected cells as control. GAPDH was used for normalization of protein content. Isoproterenol (Iso) was used as a positive control. Bars indicate average $OD_{450}$, error bars indicate SD. Values are mean ±S.D. from three independent experiments * p< 0.01, ** p<0.001, by two-way anova test. B) Relative PKA activity in HEK 293 cells transfected with wild type GPR56, GPR56 N-terminus mutant, GPR56 C-terminus mutant and GPR56 I626 and W623 double mutant. The $OD_{450}$ values corresponding to PKA activity in transfected HEK293 cells. GAPDH was used for normalization of protein content. Isoproterenol (Iso) was used as positive control. Bars indicate average $OD_{450}$, error bars indicate SD. Values are mean ±S.D. from three independent experiments * p< 0.01, ** p<0.001, by two-way anova test. C and D) Effect of testosterone on cAMP production by LNCaP and HEK 293 cells. LNCaP (c) and HEK 293(d) were treated with 10nM testosterone or isoproterenol (10 μM) for 10 min. Isoproterenol was used as positive control. cAMP levels were determined using cAMP Glo Assay. * p<0.01, **p<0.001 by two-way anova test. Mean values obtained from three independent experiments.

do not express GPR56, negligible PKA activation was observed upon testosterone treatment, while greater than two-fold PKA activation was observed in HEK293 cells overexpressing GPR56 (Fig 6B). Isoproterenol (ISO), a known activator of PKA in these cells was used as a positive control.

PKA activation was also evaluated in HEK293 cells transfected using the GPR56 N-terminal, C-terminal or double mutant. A two-folds increase in PKA activity was observed in cells transfected with C-terminal and negligible change was observed in cells transfected with N-terminal and double mutant (Fig 6B).

PKA activation is usually a result of intracellular accumulation of cAMP [32]. However, a few studies have reported cAMP-independent activation of PKA [33, 34]. Stimulation of LNCaP cells or GPR56-transfected HEK293 using 10nM testosterone, for 10 min or 20 min, resulted in significant accumulation of cAMP in these cells (Fig 6C and 6D), whereas cAMP accumulation was insignificant in GPR56 siRNA-transfected LNCaP cells or non-transfected HEK293 cells.

## 6. GPR56 induces G$\alpha_{13}$ mediated Rho activation in cells

Previous studies have revealed that GPR56 mediates its actions by activating the Rho signaling pathway via activation of G$\alpha_{12/13}$ family of G-proteins [35, 36]. To investigate whether testosterone mediated GPR56 activation also induces the Rho signaling pathway, HEK293 cells transfected with GPR56 were exposed to testosterone or vehicle alone (UT). Testosterone stimulation of GPR56-transfected HEK293 cells resulted in robust stimulation of Rho signaling pathway, as in case of positive control, whereas vehicle treatment caused negligible Rho activation (Fig 7A). Rho activation was negligible upon testosterone treatment in HEK293 cells transfected using both GPR56 and G$\alpha_{13}$ siRNA demonstrating that testosterone induced Rho activation was mediated by the G$\alpha_{13}$ pathway (Fig 7A). Also, Rho activation was unaffected when HEK293 cells were transfected with GPR56 & scrambled siRNA and treated with testosterone (Fig 7A).

## 7. Rho activation in GPR56 mutants

Rho activation assay was also performed in HEK293 cells transfected with the GPR56 N-terminal and C-terminal mutants to ascertain whether the C-terminal part of GPR56 was indeed responsible for testosterone binding and stimulation, and also whether the GPR56 C-terminal part alone, was capable of inducing Rho activation. Cells were also transfected with the double mutant to ascertain their role in testosterone binding and Rho activation. Testosterone stimulation resulted in significant Rho activation in HEK293 cells transfected with GPR56 C-terminal and GPR56, while negligible Rho activation was observed in case of HEK293 cells transfected using GPR56 N-terminus or the GPR56 double mutant (Fig 7B).

## 8. GPR56 induced PKA activation is required for Rho activation

Our results demonstrated that both PKA and Rho pathways were triggered in GPR56 expressing cells upon exposure to testosterone. PKA activation is known to promote the barrier function of cellular tight junctions, however, PKA can produce different effects based on the cell type [37]. Although cAMP/PKA seems to inhibit Rho activity in some cell types [38], yet, Leve et al have demonstrated that PKA activation promotes Rho activation and regulates the reorganization of actin cytoskeleton [37]. To investigate whether testosterone-induced Rho activation was influenced by PKA activation, HEK293 cells transfected with GPR56, GPR56 C-terminus and GPR56 N-terminus, and GPR56 double mutant were treated with PKA inhibitor H89 prior to testosterone stimulation and Rho activation assay was performed. Negligible Rho

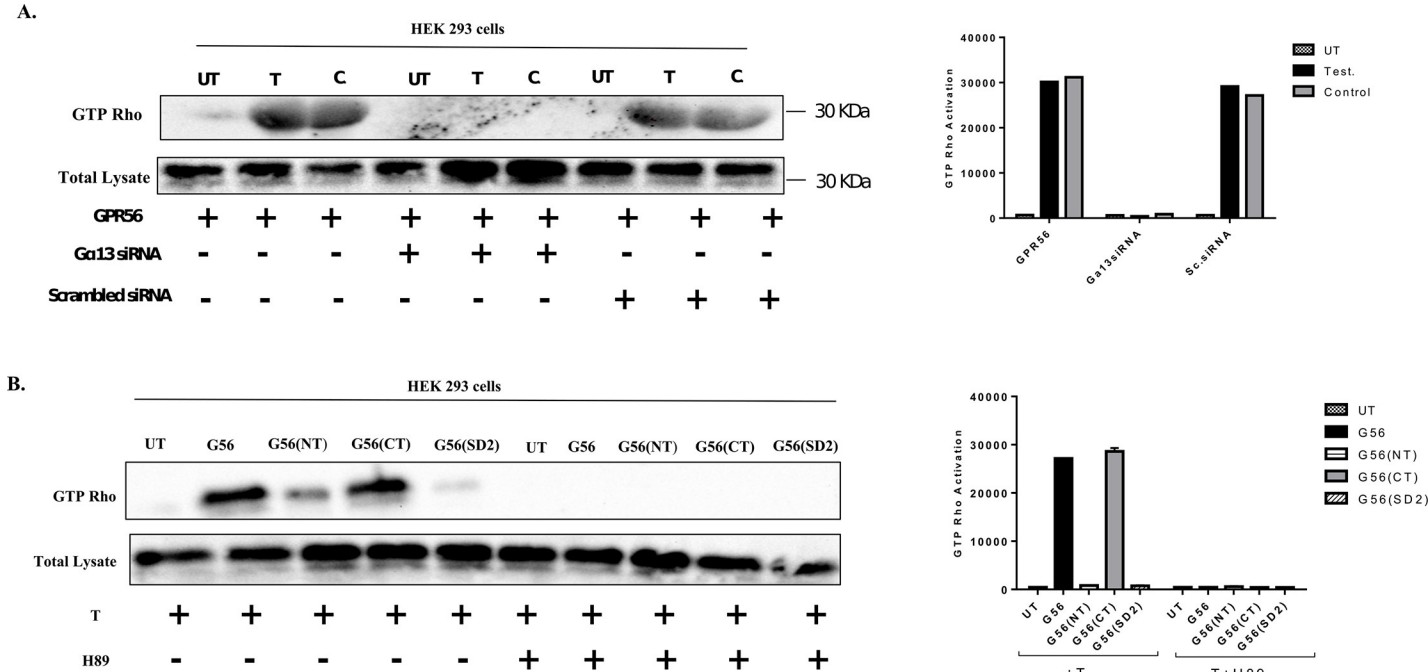

**Fig 7. Androgen binding to GPR56 triggers Rho activation.** A) HEK 293 cells were transfected using GPR56 or both GPR56 and Gα13 siRNA or GPR56 & scrambled siRNA. The cells were treated with testosterone (10nM) for 24 hrs and positive control (GTPγS) for 15 mins. The GTP-bound form of RhoA protein was detected by a pulldown assay using Rhotekin-RBD beads. Expression of RhoA in the cell lysate was estimated by western blotting using Anti-Rho antibody. Total lysate was used as the loading control. B) HEK 293 cells were transfected using GPR56, GPR56 NT mutant, GPR56 CT mutant or GPR56 double mutant and cells were treated directly with testosterone (10nM) for 24 hrs and H89 (30 μM) for 40 mins after treatment with testosterone. The GTP-bound form of RhoA protein was detected by a pulldown assay using Rhotekin-RBD beads. Expression of RhoA in the cell lysate was estimated by western blotting using anti-Rho antibody. Total lysate was used as the loading control.

activation was observed in H89 treated cells indicating that testosterone induced PKA activation was important for Rho activation in these cells (Fig 7B).

## 9. GPR56 regulates the subcellular localization of AR

In prostate cells testosterone exposure induces the translocation of AR from cytosol to nucleus. In LNCaP cells, transfected with GFP-tagged AR, most of the GFP-AR had translocated into the nucleus within 1 hour (Fig 8A). However, in LNCaP cells where GPR56 expression was knocked down using GPR56-siRNA, partial migration of GFP-AR was observed in the cytoplasm and nucleus after 1 hour (Fig 8B). Complete nuclear translocation of GFP-AR did not occur in these cells till 24 hours of testosterone treatment (S3 Fig). To confirm that nuclear translocation of AR was indeed impaired in GPR56 siRNA transfected LNCaP cells, western blot analysis was performed using nuclear and cytoplasmic fractions of LNCaP cells transfected with scrambled siRNA or GPR56 siRNA and treated with vehicle alone or testosterone (Fig 8C). The results demonstrate that in LNCaP cells expressing GPR56, the AR is in cytoplasm in vehicle treated cells, but moves to the nucleus on testosterone treatment, whereas in the cells transfected using GPR56 siRNA, upon testosterone stimulation a significant proportion of AR continues to reside in the cytoplasm and does not migrate into the nucleus (Fig 8C).

## 10. GPR56 induces cell growth and is over expressed in tumor samples

GPR56 has been proposed to play varying roles in tumor cell proliferation and metastasis [39–42]. However, whether it accelerates the growth of tumor cells or not is still ambiguous. The

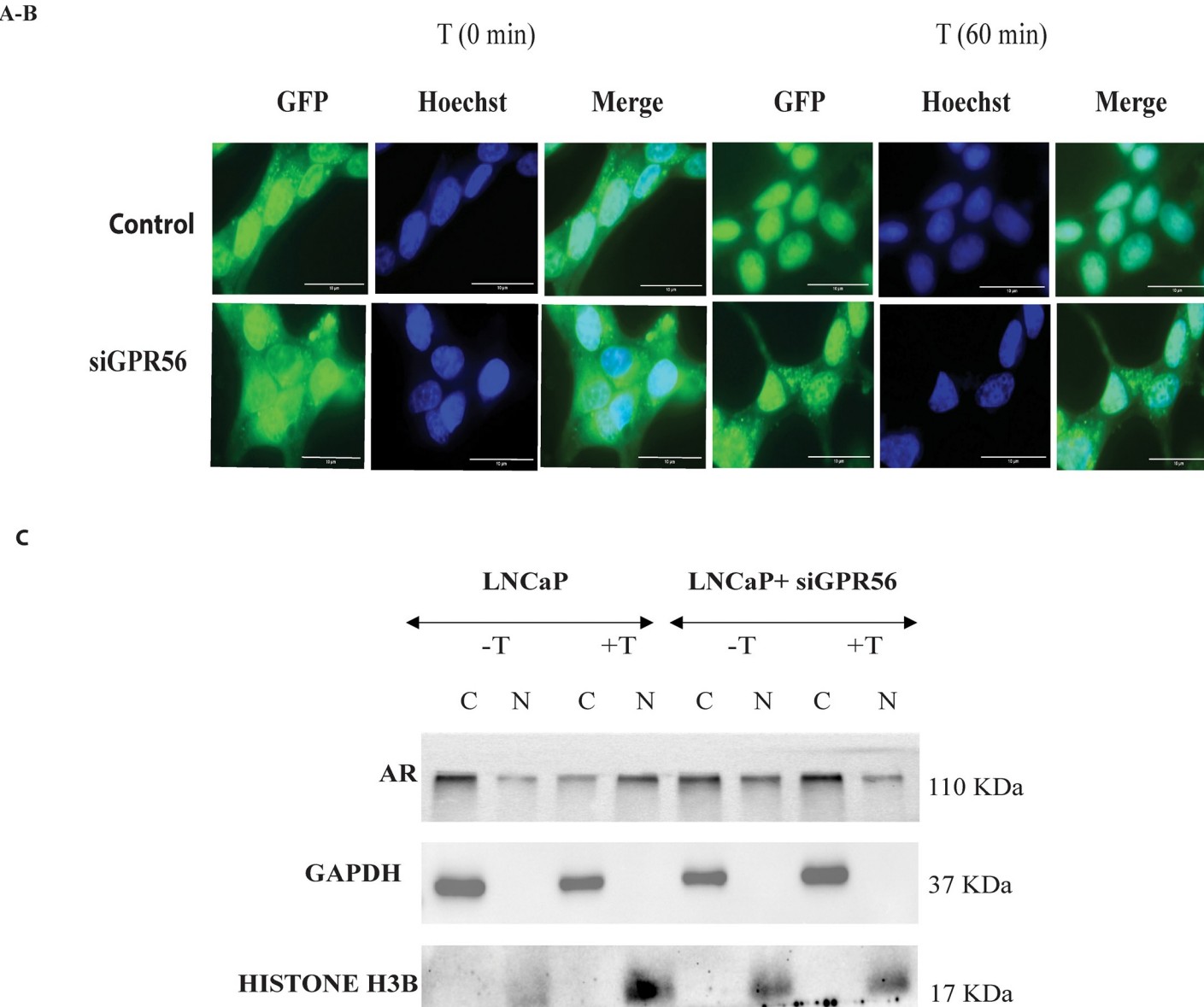

**Fig 8. GPR56 regulates the subcellular localization of AR.** A-B) Knockdown of GPR56 expression causes inhibition of AR translocation: LNCaP cells were transfected using GFP-AR, or GFP-AR and SiGPR56 (100nM), and treated with 10 nM testosterone for 24 h after transfection. Fluorescence Images captured at various time intervals i.e. 5 mins, 15 mins, 30 mins, 1 hour, 4 hours, 24 hours (other time intervals are shown in S2 Fig). Nuclei were visualized by Hoechst staining. Scale bar, 50um (added using Image J). C) Cytoplasmic and nuclear fractionation of AR in unstimulated (vehicle treated) and testosterone stimulated cells in presence and absence of GPR56 siRNA (siGPR56). LNCaP cells transfected with siRNA against GPR56 (120nM) were treated using vehicle or testosterone (10nM) for 1 hours. Western blotting was done using 50 ug of protein from cells lysate using AR antibody. Glyceraldehyde-3-phosphate dehydrogenase (GAPDH) antibody was used as a cytoplasmic fraction marker and Histone H3B antibody as a nuclear fraction marker.

MTT assay was performed in LNCaP and HEK293 cells to assess the role of GPR56 in cell proliferation. The GPR56 siRNA transfected LNCaP cells, expressing negligible GPR56, exhibited significantly less cell growth compared to normal LNCaP cells, while overexpression of GPR56 in HEK293 cells increased cell proliferation compared to non-transfected HEK293 cells (Fig 9A and 9B). Also, GPR56 siRNA transfected PC3 cells, expressing negligible GPR56, displayed significantly less cell proliferation compared to normal PC3 cells (Fig 9C)

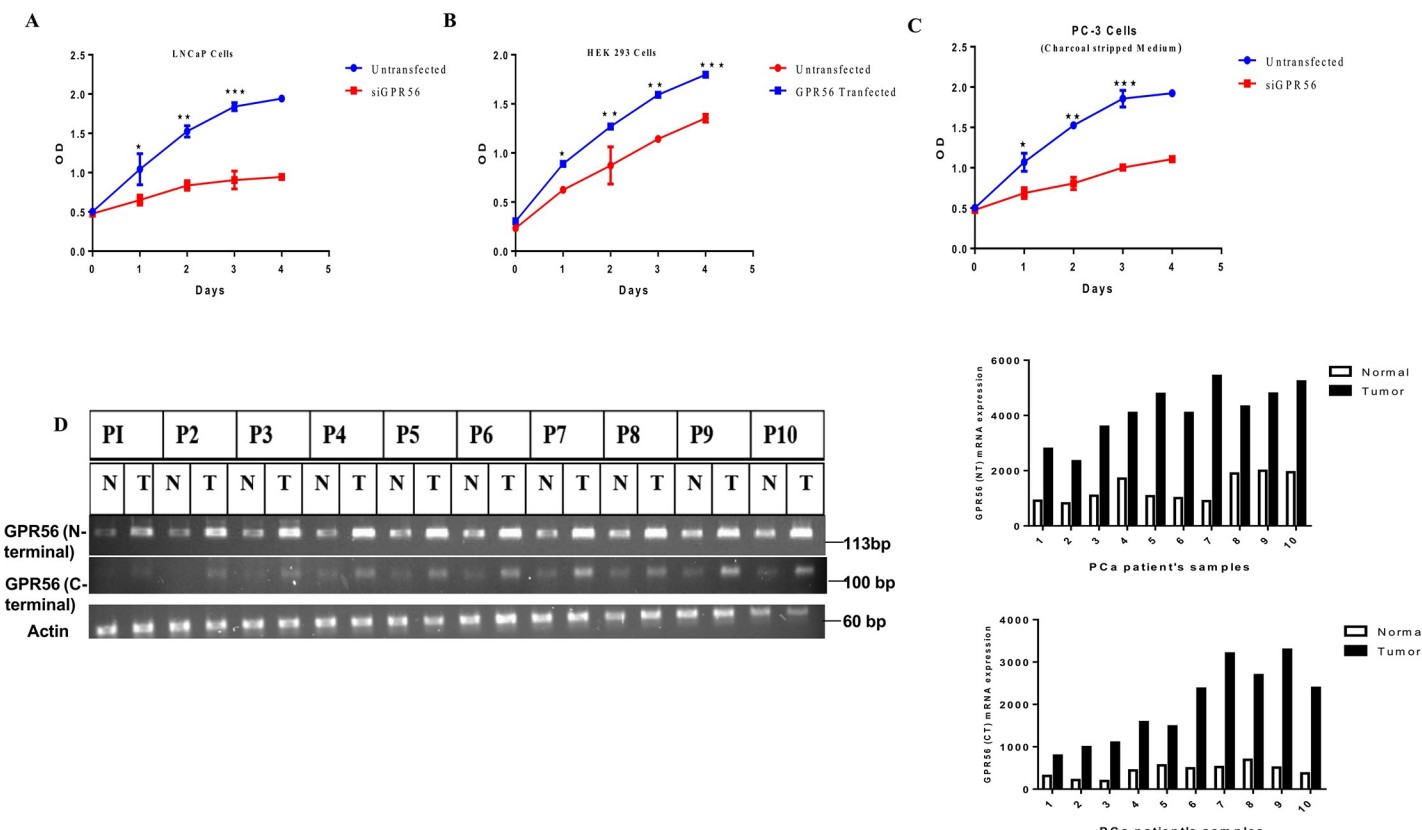

**Fig 9. GPR56 expression in tumor samples.** A B & C) MTT assay in LNCaP, HEK 293 and PC3 cells. Cell proliferation in LNCaP cells transfected with GPR56-siRNA. Cell proliferation in LNCaP cells and cells transfected with siGPR56 were examined at 24 h, 48 h, 72 h or 96 h after seeding, using MTT assay. The data represents the mean ±S.D. of three independent experiments. Values are mean ±S.D. from three independent experiments. * p< 0.01, ** p<0.001, *** p<0.0001, two-way anova test B) Cell proliferation in non-transfected and GPR56 transfected HEK cells was examined 1–4 days after seeding using MTT assay. The data represents the mean ±S.D. of three independent experiments. * p< 0.01, ** p<0.001, *** p<0.0001, two-way anova test. Cell proliferation in PC3 cells transfected with GPR56-siRNA. Cell proliferation in PC3 cells and cells transfected with siGPR56 were examined at 24 h, 48 h, 72 h or 96 h after seeding, using MTT assay. The data represents the mean ±S. D. of three independent experiments. Values are mean ±S.D. from three independent experiments. * p< 0.01, ** p<0.001, *** p<0.0001, two-way anova test D) Representative RT-PCR analysis of GPR56 in matched normal (N) vs prostate tumor (T) tissue from individual patient's samples (P1-P10) (P11-P25) patient's samples. RNA was isolated from prostate tumor (T) and matched normal (N) tissue from individual patients and reverse transcribed using gene specific primers for GPR56 (as described in materials and methods). The amplified products were resolved on 2% agarose gel. The bands corresponding to GPR56 (N- terminal)-143 bp and GPR56 (C-terminal)-110 bp were observed. Beta actin was used as control. Quantification of band densities for N-terminal or the C- terminal region of the GPR56 mRNA transcript in patient's tissue samples has been shown (using Image J).

To investigate whether there is any difference in GPR56 expression in normal versus tumor samples RT-PCR was performed. RNA was isolated from paired normal and tumor prostate tissue samples from 25 patients. Using two sets of GPR56 primers which encompassed parts of either the N-terminal or the C-terminal regions of the GPR56 mRNA transcript, keeping in view the large number of possible GPR56 transcripts in a cell. GPR56 expression was observed to be higher in all tumor samples examined as compared to the normal samples. When using both the N-terminal and C-terminal primer sets (Fig 9D). Quantification of band densities for N-terminal or the C- terminal region of the GPR56 mRNA transcript in patient's tissue samples has been shown. To demonstrate the changes in GPR56 expression at the protein level in normal versus tumor tissue, western blot analysis has been performed using GPR56 antibody (Fig 10A) and the intensity quantification is depicted, the expression of GPR56 is significantly higher at the protein level in all tumor tissue analyzed. TCGA data analysis using gene expression data from 495 tumor and normal samples from GDC portal of National Cancer Institute

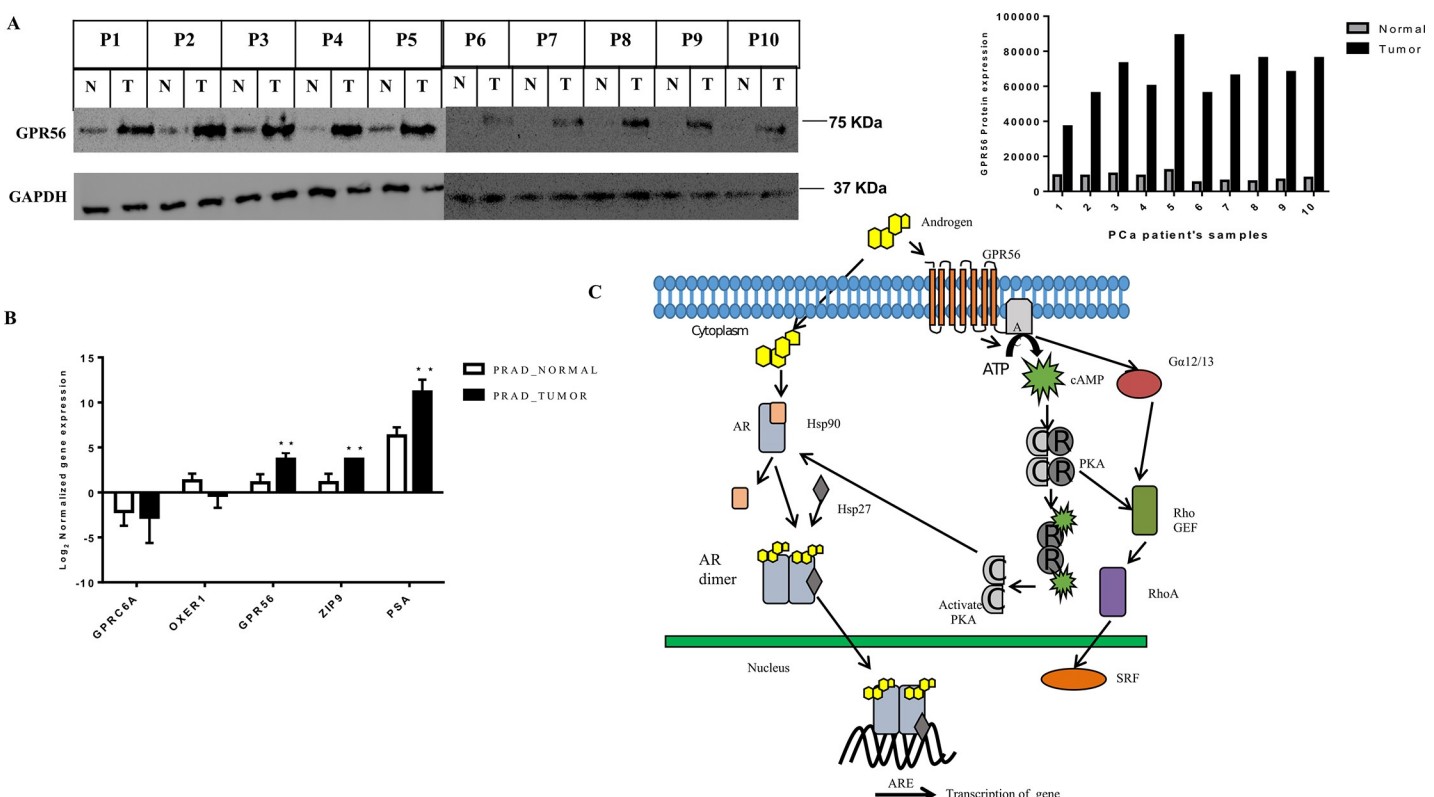

**Fig 10.** A) GPR56 protein expression in normal versus tumor tissue. Western blot analysis has been performed using GPR56 antibody. Western blotting was done using 50 ug of protein from tissue lysate using GPR56 antibody. Glyceraldehyde-3-phosphate dehydrogenase (GAPDH) was used as a loading control. GPR56 protein band intensity quantification in normal versus tumor tissue was done using Image J software. B) Analysis of TCGA data of Membrane androgen receptors genes (namely GPR56, GPRC6A, ZIP9, OXER1) expression and comparison with PSA gene expression in Normal Prostate (45) Vs Prostate Tumor patient's samples (450) (TCGA_Prad). The analysis is done using GDC portal for cancer genomics. C). Following the classical pathway, testosterone transverses the membrane to enter the cytosol and bind to AR. It also activates GPR56 present on the membrane and causes activation of PKA and Rho pathway by non-genomic mechanism. Activation of both pathways are necessary for the translocation of AR into the nucleus and transcription of AR target genes.

(http://portal.gdc.cancer.gov) revealed almost 3.4 fold higher expression of GPR56 in tumor samples as compared to normal samples indicating an overall difference in expression of GPR56 in prostate cancer patients (Fig 10B).

## Discussion

Endocrine therapies aimed at inhibiting AR function comprise the primary treatment plan for PCa patients. However, the tumor reappears in a more aggressive form due to the activation of alternate cellular pathways which directly or indirectly activate the AR and promote cell proliferation. Identification of novel signaling molecules/ pathways that may play a critical role in aberrant AR activation in prostate, is of utmost importance for developing new treatment strategies and drug targets for patients with advanced prostate cancer. Here we demonstrate androgen-mediated activation of adhesion GPCR, GPR56, that contributes to classical AR signaling in prostate cells. Activation of GPR56 by other ligands, in a low/no androgen situation, as in case of CRPC, may activate downstream pathways and cause aberrant activation of the AR.

A simple bioinformatics-based approach was used to identify GPCRs which exhibit maximum sequence similarity with the LBD of the AR. In spite of limitations associated with this approach, in absence of available crystal structures, structures were generated *in silico* and

docking studies were performed to evaluate the possible interaction of these GPCRs with androgens. Binding of testosterone to GPR56 was most favorable energetically (-8.32 Kcal/mol) as compared to other GPCRs such as GPRC6A (-5.886 Kcal/mol) or GPR663 (-5.011 Kcal/mol) which have been reported earlier. The energy of minimization value was slightly higher than that of nuclear AR(-12.3Kcal/mol) probably due to presence of more interacting residues in the androgen binding cleft of nuclear AR (primarily Asn 705, Thr 877 on top and arg 752 at the bottom) and the geometry of AR-androgen binding cleft. The important interacting residues for androgen in case of GPR56 are Ile 626, trp 623 and trp 563 which were also validated by site directed mutagenesis experiments. GPR56 was therefore taken up for further studies. The predicted binding site for testosterone is towards the C-terminus of the receptor, whereas that of collagen III and TG2, two known ligands of GPR56, is towards the N-terminus [42, 43].

In our studies binding of testosterone to GPR56 was assessed by flow cytometry in prostate cell lines such as LNCaP which express GPR56. Binding specificity was established by the fact that knocking down the GPR56 expression in LNCaP cells by transfecting GPR56 siRNA, significantly reduced binding of T-BSA-FITC to these cells. To confirm that testosterone actually bound to the C-terminal domain of GPR56, a C-terminal clone that included only the residues predicted to bind to testosterone (511–695) and an N-terminal construct (1–509) including the entire protein excluding the testosterone binding domain were generated (Fig 3A). Interestingly, only the C-terminal fragment and GPR56 exhibited strong binding to T-BSA-FITC while the N-terminal construct representing the N-terminal part, GAIN domain and most of the 7TM domain, exhibited negligible binding. Also, the double mutant in which two critical testosterone binding residues, namely, Ile 626 and Trp 623 had been altered showed negligible binding to testosterone confirming the key role of these residues in testosterone binding.

The Multiple sequence alignment of Full length GPR56 which reveals the conservation of critical Trp residues is in agreement with earlier analysis by Graaf et al, [44] and points to the importance of these residues in GPR56 function. Also, our study provides the first example of a ligand binding to the C-terminal domain of GPR56. Earlier studies by Pi et al have demonstrated that testosterone binds towards the C-terminal end of GPRC6A, another membrane GPCR [45]. The critical residues of GPR56 as well as GPRC6A, predicted to bind to testosterone, belong to transmembrane domains, TM5 and TM6 [45]. The TM6 of the 7TM plays an important role in receptor activation, as it moves away from the transmembrane bundle to aid G-protein activities [46]. However, as the Ile and Trp residues, predicted to interact with testosterone are included in TM6, it would be interesting to investigate the mechanism of testosterone binding and subsequent receptor activation.

Studies by Stoveken et al have demonstrated that GPR56 undergoes cleavage at the GAIN domain and the seven trans-membrane region is important for the activity of the receptor [35]. However, the C-terminal mutant in our studies comprising a small part of the entire 7TM domain was capable of testosterone binding and stimulation indicating that the entire 7TM domain was not required for eliciting response. The double mutant, in which only two critical testosterone binding residues had been altered was deficient in testosterone binding and demonstrated negligible activation of the downstream pathways. Whether testosterone binding also leads to GAIN domain cleavage or not, awaits further investigation. Testosterone-stimulation of GPR56 caused downstream activation of both PKA and Rho signaling pathways. Activation of G13-mediated Rho signaling is a typical signaling route induced by GPR56 ligands [28]. Although testosterone did not bind to the N-terminal of GPR56, yet, it stimulated the Rho pathway like the other GPR56 ligands. Notably, PKA activation was necessary for Rho activation by testosterone as inhibiting PKA activation by H89 inhibited Rho activation.

Binding of testosterone to GPRC6A also activates both PKA and MAP kinase signaling pathways and PKA activation is necessary for testosterone- induced ERK activation [45].

Though cAMP and PKA activation are mostly a consequence of activation of G$\alpha$s signaling axis, yet a few instances have revealed G$_{13}$ mediated activation of cAMP through adenylate cyclase 7 [47]. Whether the cAMP/PKA activation by testosterone-stimulation of GPR56 is G$_{\alpha s}$ or G$_{13}$ mediated would require detailed studies.

Previous studies have shown that PKA activation is a prerequisite step for phosphorylation of HSP90 bound to unliganded AR. Phosphorylation of HSP90 causes release of AR and its nuclear migration [31]. Knocking down the expression of GPR56 or PKA in LNCaP cells using respective siRNA reduced the AR transcription significantly. Also, in HEK 293 cells transfecting GPR56 along with AR enhances transcription, AR transcription is minimal when these cells are co-transfected with GPR56 N-terminal or GPR56 double mutant.

Also, as expected, knockdown of GPR56 expression, led to sequestration of AR in the cytoplasm. While most of the AR had translocated into the nucleus within 1 hour in LNCaP cells, the import of GFP-AR in cells transfected with GPR56 siRNA took much longer and was only partial even after 24 hours of treatment. Accordingly, reduction in ARE-regulated reporter gene activity as well as expression of AR target genes such as PSA and TMPRSS2 was observed in GPR56-siRNA transfected LNCaP cells.

Based on our findings and earlier reports existence of parallel androgen effector pathways seem to be active in cells. A non-genomic pathway mediated via GPR56 (and possibly other GPCRs) that activates cAMP and PKA and a genomic pathway that is mediated through the nuclear AR by direct binding of androgens (Fig 10C). Activation of both these pathways is required for complete androgen signaling. Thus, activation of GPCRs such as GPR56 by androgens is necessary for activating the PKA pathway that uncouples the AR from HSP90, so that it can migrate into the nucleus to execute transcription modulation of target genes and cell proliferation.

The physiological role of androgen binding to GPR56 is still not clear. Although much is known about GPR56 function in brain, yet, comparatively little is revealed about its role in organs such as the prostate. Some of the effects of androgens in brain and other organs could be mediated by GPR56. Androgens play a critical role in neuron myelination in mice and castration impairs recruitment of astrocytes to axons [48]. GPR56 is a regulator of oligodendrocyte development and regulating myelin sheath formation the effect of androgens on myelin sheath formation could be mediated by GPR56 [49]. Overexpression of the C-terminal domain of GPR56 causes increased secretion of VEGF in melanoma cells [50]. Androgens enhance VEGF expression both at the RNA and protein level [51]. Whether androgen binding to the C-terminal domain of GPR56, plays a role in increased VEGF secretion needs further investigation.

Our study demonstrates that extensive positive cross talk exists between the genomic and non-genomic mode of AR signaling. While androgens activate GPR56 resulting in activation of PKA and Rho signaling pathways, activated PKA in turn is necessary for classical AR signaling. In CRPC patients, who have low concentrations of androgen, aberrant activation of GPR56 or PKA inside the cell by agents other than androgens can trigger AR activation [52]. This could be a major reason for the failure of endocrine therapies in CRPC patients [3]. Of the twenty-five prostate tumor samples that we investigated, GPR56 mRNA expression was higher in all samples tested revealing its possible involvement in tumor cell proliferation. To validate, two different sets of primers were used for analyzing GPR56 expression in patient samples, in view of the large number of GPR56 transcripts of various sizes observed in cells (www.ncbi.nlm.nih.gov/gene/9289). However, though the sample size was low, yet all patient samples displayed an increase in GPR56 mRNA expression using both primer sets

(encompassing a part of either N-terminal or C-terminal domain). Similarly, GPR56 protein expression was also significantly higher in all tumor samples compared to normal. TCGA data analysis using gene expression data from 495 tumor and normal samples revealed almost 3.4-fold higher expression of GPR56 in tumor samples as compared to normal samples indicating an overall difference in expression of GPR56 in prostate cancer patients.

The role of GPR56 as a membrane androgen receptor is strengthened by the fact that mice lacking GPR56 expression (GPR56 -/-) show impaired or no testes development [52]. The precise role of GPR56 in male reproduction and prostate cancer development needs to be established. Currently, GPCRs represent the single largest drug target as they recognize a wide variety of ligands and are found almost ubiquitously in the body [53]. Development of antagonists and drug targets against GPR56 may be effective in limiting prostate tumor growth, however such targets need to be designed carefully, so that the positive role of GPR56 in brain development is not compromised. Further studies are required to completely elucidate the role of GPR56 as a membrane androgen receptor and its function in different organs of the body.

## Summary

In this study, GPR56, an adhesion GPCR, was identified and established as an androgen-binding GPCR. Testosterone-binding of GPR56 results in activation of PKA and Rho signaling pathways. This GPR56-induced activation of PKA promotes AR signaling indirectly by causing the nuclear translocation and of AR, and subsequent cell proliferation. Thus, activation of GPR56-mediated signaling contributes to genomic signaling by AR. GPR56 expression is increased significantly in prostate tumor samples compared to normal tissue, at both RNA and protein levels. This is also verified by analysis of TCGA data of 495 PCa patients in the database. Thus, in low/no androgen conditions, increased GPR56 expression and activation by other agents may lead to continued AR signaling as observed in CRPC. Our study suggests a role for GPR56 in androgen signaling and PCa and may provide an additional target for PCa treatment in the future.

## Supporting information

**S1 Table. Clinical characteristics of 25 PCa patients.**
(DOCX)

**S1 Fig. Phylogenetic tree was constructed to study the evolutionary relationships among primary amino acid sequences of the target GPCRs using clustal omega.** The numbers in italics are the bootstrap values. Putative membrane ARs (GPRC6A, ZIP9 and GPCR 663) proposed in other studies were also included to assess the relatedness among their sequences.
(TIF)

**S2 Fig. Androgen Receptor transactivation through GPR56 in charcoal stripped medium.** A) Inhibition of AR transcription by knocking down GPR56 and PKA expression using charcoal stripped medium. LNCaP cells transfected with ARE-Luc (1μg) reporter plasmid with scrambled siRNA(100nM) or siGPR56 (100nM) and siPKA (120nM). The cells were treated with 0.1 and 10nM testosterone 24 h after transfection period. Values are mean ±S.D. from three independent experiments. * p< 0.01, ** p<0.001, two-way anova test. B) AR transcription analysis of GPR56-NT, GPR56-CT, and GPR56 double mutant. HEK293 cells transfected with ARE-Luc (1μg) reporter plasmid with GPR56, GPR56 N terminus mutant, GPR56 C terminus mutant, GPR56 double mutant (SD2) plasmids using charcoal stripped medium. The cells were treated with 10nM testosterone 24 h after transfection period. Values are mean ±S. D. from three independent experiments. * p< 0.01, ** p<0.001, two-way anova test.C) Cell

proliferation in PC3 cells transfected with GPR56-siRNA. Cell proliferation in PC3 cells and cells transfected with siGPR56 were examined at 24 h, 48 h, 72 h or 96 h after seeding, using MTT assay performed in charcoal stripped medium. The data represents the mean ±S.D. of three independent experiments. Values are mean ±S.D. from three independent experiments. $^*$ p< 0.01, $^{**}$ p<0.001, $^{***}$ p<0.0001, two-way anova test.
(TIF)

**S3 Fig. GPR56 regulates the subcellular localization of AR.** A-B) Knockdown of GPR56 expression causes inhibition of AR translocation: LNCaP cells were transfected using GFP-AR, or GFP-AR along with siGPR56 (100nM), and treated with 10 nM T 24 h after transfection. Fluorescence Images captured at various time intervals 5 mins, 15 mins, 30 mins, 1 hour, 4 hours, 24 hours. Nuclei were visualized by Hoechst staining. Scale bar,10um (added using Image J).
(TIF)

**S1 Raw images.**
(PDF)

**S2 Raw images.**
(PDF)

## Acknowledgments

The authors would like to thank Professor. Yehia Daaka, University of Florida for his valuable inputs. We thank Dr. Hsi-Hsein Lin, Director, Graduate Program of molecular medicine, Department of Microbiology and Immunology, College of Medicine, Chang Gung University, Taiwan for providing plasmid for GPR56. The authors like to thank Dr. Pradeep Kumar Rai, Application Scientist, BD-JH (FACS Academy), Jamia Hamdard, Delhi, India for providing technical support in Flow cytometry experiments. The authors also like to thank Dr. Sandeep Saxena, National Institute of Immunology, Delhi, India for kind gift of Histone H3B antibody.

## Author Contributions

**Conceptualization:** Gargi Bagchi.

**Data curation:** Julie Pratibha Singh, Ravi Datta Sharma.

**Formal analysis:** Julie Pratibha Singh, Ravi Datta Sharma.

**Investigation:** Julie Pratibha Singh, Manisha Dagar, Gunjan Dagar, Sudhir Kumar, Gargi Bagchi.

**Resources:** Sudhir Rawal, Ravi Datta Sharma, Rakesh Kumar Tyagi.

**Supervision:** Gargi Bagchi.

**Writing – original draft:** Gargi Bagchi.

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
