## [Decision Letter · Decision Letter 0]

14 Jan 2020

PONE-D-19-31997

Activation of GPR56, a novel adhesion GPCR, is necessary for nuclear androgen receptor signaling in prostate cells

PLOS ONE

Dear Dr. Gargi Bagchi:

Thank you for submitting your manuscript to PLOS ONE. After careful consideration, we feel that it has merit but does not fully meet PLOS ONE’s publication criteria as it currently stands. Therefore, we invite you to submit a revised version of the manuscript that addresses the points raised during the review process.

**Reviewer #1: **

Singh and coworkers conducted a study entitled “Activation of GPR56, a novel adhesion GPCR, is necessary for nuclear androgen receptor signaling in prostate cells” in which the authors indicated that androgens activate Rho-signaling pathway through the activation of GPR56. GPR56 knockdown blocks the nuclear translocation of AR and its downstream transcription factors. To validate the in vitro studies, GPR56 was upregulated in prostate cancer (PCa) tumor specimens when compared with normal tissues.

General comments:

The manuscript was well-designed, technical aspects used in this study are appropriate, results are supported with enough data and the size of the text is reasonable. However, I have some few comments need be addressed in the current study.

- Using LNCaP as a model is appropriate here but PC-3 representing CRPC cells should be also included as a control.

- Examining the expression of GPR56 transcript in prostate cancer tissue is not enough because posttranslational modification can change the whole story. It is better to stain the tissue with GPR56 specific antibody. In addition, the authors showed that GPR56 triggers cytosolic translocation of AR using immunofluorescence, and it would be more believable if the authors use cell fractionation to validate the results.

Specific comments:

- In the abstract: Line 4, the phrase of “a role for these in CRPC” should be fixed. Please include bioinformatics results of AR-ligand similarity and the number of examined PCa tissues (25 tissue samples) in the abstract.

- In Methods section, please replace (u: micron) with Latin symbol (µ). Clinical characteristics of human prostate cancer samples are needed.

- In results section, all figures need an improvement because of quality was very poor. Figure legends must be included together after the MS text not included in the results. In results, page 14, please replace S 1 Fig. with Supl. Fig. 1. Statistical analysis should be included in Figure 7A&B. In Fig. 7C, quantification of band densities will improve the results of GRP56 in tissues samples.

- In discussion section, page 21, “Cancerous prostate” should be replaced. I suggest to move Figure 8 A&C and also Tables 3&4 to the results section (The Authors should discuss the results but not to include new results). At the end of discussion, summary or conclusion should be include in which you summarize your findings. Discussion should be more concise, focus and comprehensive.

- The manuscript should be edited again because it has several typos and incorrect structures.

**Reviewer #2: **

In the current manuscript titled “Activation of GPR56, a novel adhesion GPCR, is necessary for nuclear androgen receptor signaling in prostate cells” demonstrates the role of GPR56 in regulating AR downstream signaling. The authors have demonstrated how binding of the testosterone to the GPR56 can activate PKA and Rho signaling promoting AR nuclear translocation demonstrating crosstalk between two signaling pathways. However, following concerns need to be addressed prior to publication.

1- Since CRPC occurs in patients having very low levels of testosterone it does not explain how GPR56 can lead to CRPC which the authors are claiming (the docking score for AR is much higher for testosterone than GPR56, table 2) this needs to be clarified in the discussion.

2- AR receptor signaling studies require the testosterone induction to be performed in phenol red free charcoal stripped serum media, authors have used serum starvation instead that is not ideal for these studies.

3- The authors need to demonstrate if GPR56 siRNA can block AR nuclear localization using cell fractionation studies.

4- Fig3H-The PSA levels go down after testosterone treatment whereas GPR56 expression go up. Testosterone upregulates AR signaling and PSA expression, authors need to explain this abnormal result. Also they need to show effect on another AR regulated gene.

Minor concerns:

1- Authors have included the figure legends in between the manuscript text without the figure that is not reader friendly.

2- Constructs- is it at the HindIII site or between HindIII and another restriction enzyme?

3- The GTP Rho pull down experiment not described properly in methods section.

4- Was the cyclic AMP assay performed 2-4 hrs after serum starvation or after 24 hrs of starvation?

5- Page 15, correct that GPR56 protein is not expressed in HEK cells

6- Fig 3F-Is the control scrambled siRNA?

7- Rephrase “androgen transcription” to AR downstream signaling, authors are not looking into AR mRNA levels.

8- Fig4A-PKAsiRNA set is still showing response to ISO.

9- Fig 7- provide a western blotting with N and C terminal GPR56 antibodies.

We would appreciate receiving your revised manuscript by Feb 28 2020 11:59PM. To enhance the reproducibility of your results, we recommend that if applicable you deposit your laboratory protocols in protocols.io, where a protocol can be assigned its own identifier (DOI) such that it can be cited independently in the future. For instructions see: http://journals.plos.org/plosone/s/submission-guidelines#loc-laboratory-protocols

We look forward to receiving your revised manuscript.

Kind regards,

Mohammad Saleem

University of Minnesota
---

## [Author Response · Author response to Decision Letter 0]

4 Mar 2020

Dear Editor,

Thank you for considering our article. Also, thanks to the reviewers who have read the manuscript meticulously and provided constructive comments. According to the reviewers’ suggestions, we have now performed more experiments and have included them in the manuscript text and figures. The pointwise response is also included below.

Reviewer #1: 

Singh and coworkers conducted a study entitled “Activation of GPR56, a novel adhesion GPCR, is necessary for nuclear androgen receptor signaling in prostate cells” in which the authors indicated that androgens activate Rho-signaling pathway through the activation of GPR56. GPR56 knockdown blocks the nuclear translocation of AR and its downstream transcription factors. To validate the in vitro studies, GPR56 was upregulated in prostate cancer (PCa) tumor specimens when compared with normal tissues.

General comments:

The manuscript was well-designed, technical aspects used in this study are appropriate, results are supported with enough data and the size of the text is reasonable. However, I have some few comments need be addressed in the current study.

1. - Using LNCaP as a model is appropriate here but PC-3 representing CRPC cells should be also included as a control.

Response: The PC3 cell line has now been used along with LNCaP cells for performing Luciferase assay and MTT assay and the results have now been included in the text and figures ( 3H and 8C). It is important to note that the response in PC3 cells is similar to that of LNCaP.

2. Examining the expression of GPR56 transcript in prostate cancer tissue is not enough because posttranslational modification can change the whole story. It is better to stain the tissue with GPR56 specific antibody. 

Response: To demonstrate the changes in GPR56 expression at the protein level in normal versus tumour tissues, Western Blot analysis has been performed using normal and tumour tissues using GPR56 antibody. The result is now included as Figure 8A and the intensity quantitation is depicted as Fig. 8B. The expression of GPR56 is significantly higher at the protein level in all tumour tissues analysed.

3.In addition, the authors showed that GPR56 triggers cytosolic translocation of AR using immunofluorescence, and it would be more believable if the authors use cell fractionation to validate the results.

Response: As per the reviewer’s suggestion, we have now performed Western Blot experiment using nuclear and cytoplasmic fractions of LNCaP cells transfected using scrambled siRNA or GPR56 siRNA, and probed using anti-AR antibody and the results have been included as Fig. 6C. The results clearly demonstrate that in LNCaP cells expressing GPR56, the AR is in the cytoplasm in untreated cells, but moves to the nucleus on testosterone treatment, whereas in the cells transfected using GPR56 siRNA, upon testosterone stimulation, a significant proportion of AR continues to reside in the cytoplasm and does not migrate into the nucleus.

Specific comments:

- In the abstract: Line 4, the phrase of “a role for these in CRPC” should be fixed. 

Response: This sentence has now been corrected.

Please include bioinformatics results of AR-ligand similarity and the number of examined PCa tissues (25 tissue samples) in the abstract.

Response: The details have now been included in the Abstract.

- In Methods section, please replace (u: micron) with Latin symbol (µ). 

Response: Corrected

Clinical characteristics of human prostate cancer samples are needed.

Response: Clinical Characteristics of patients have now been included in the M and M section as Supplementary Table 1.

- In results section, all figures need an improvement because of quality was very poor. 

Response: High resolution images for all figures have now been included in Tiff format 300 Dpi.

Figure legends must be included together after the MS text not included in the results.

Response: Figure legends have now been included after the MS text.

 In results, page 14, please replace S 1 Fig. with Supl. Fig. 1.

Response: Corrected

 Statistical analysis should be included in Figure 7A&B. 

In Fig. 7C, quantification of band densities will improve the results of GRP56 in tissues samples.

Response: Statistical Analysis has now been included for Fig. 7A, B and C and quantification of band densities has also been included for Fig. 7D.

- In discussion section, page 21, “Cancerous prostate” should be replaced. 

Response: Has been replaced.

I suggest to move Figure 8 A&C and also Tables 3&4 to the results section (The Authors should discuss the results but not to include new results). 

Response: Figures 8A and C have now been moved to Results section as Fig. 3A and Fig. 8C.

Tables 3 and 4: Have now been moved to Results Section

At the end of discussion, summary or conclusion should be include in which you summarize your findings. 

Response: A Summary section has now been included.

Discussion should be more concise, focus and comprehensive.

- The manuscript should be edited again because it has several typos and incorrect structures.

Response: We have edited the MS again to correct typos.

Reviewer #2: 

In the current manuscript titled “Activation of GPR56, a novel adhesion GPCR, is necessary for nuclear androgen receptor signaling in prostate cells” demonstrates the role of GPR56 in regulating AR downstream signaling. The authors have demonstrated how binding of the testosterone to the GPR56 can activate PKA and Rho signaling promoting AR nuclear translocation demonstrating crosstalk between two signaling pathways. However, following concerns need to be addressed prior to publication.

1- Since CRPC occurs in patients having very low levels of testosterone it does not explain how GPR56 can lead to CRPC which the authors are claiming (the docking score for AR is much higher for testosterone than GPR56, table 2) this needs to be clarified in the discussion.

Response: While in CRPC testosterone levels are reduced, yet the expression of most GPCRs and their ligands are significantly enhanced (Daaka, 2005). Increase in expression of GPR56 in tumour samples over their normal counterparts has been demonstrated in this study (Fig.7 and 8) and through TCGA analysis of previous studies (Fig. 8C). Increased expression of GPR56 via any means is likely to result in increased activation of PKA, which would result in increased nuclear migration of AR due to phosphorylation of HSP90 (Dagar et al, 2019). This would result in enhanced cell proliferation in low/no testosterone conditions as observed in CRPC. This has now been explained more clearly in Discussion section in _______

2- AR receptor signaling studies require the testosterone induction to be performed in phenol red free charcoal stripped serum media, authors have used serum starvation instead that is not ideal for these studies.

Response: In literature, some studies have used serum starvation while others have used charcoal stripped serum. We repeated few of the experiments in Charcoal stripped serum to verify that the results were similar to those obtained with serum starvation. These results are now included as Supplementary Figure 2.

3- The authors need to demonstrate if GPR56 siRNA can block AR nuclear localization using cell fractionation studies.

Response: This has now been included in the text and Results section as Figure 6C.

4- Fig3H-The PSA levels go down after testosterone treatment whereas GPR56 expression go up. Testosterone upregulates AR signaling and PSA expression, authors need to explain this abnormal result. Also they need to show effect on another AR regulated gene.

Response: To verify the effect of GPR56 knockdown, LNCaP cells were transfected with GPR56 siRNA or scrambled and expression levels of genes such as PSA and GPR56 was analysed. 

This experiment has now been repeated to verify the results and expression of both PSA and GPR56 are upregulated upon Testosterone treatment. Expression of another androgen target gene, namely TMPRSS2 was also analysed and found to increase upon Testosterone treatment. Expression of all the three genes were negligible in LNCaP cells transfected using GPR56 siRNA.

This has now been included in text, page number 16.

Minor concerns:

1- Authors have included the figure legends in between the manuscript text without the figure that is not reader friendly.

Response: Figure legends have now been included at the end of the text.

2- Constructs- is it at the HindIII site or between HindIII and another restriction enzyme?

Response: It is at the Hind III site and has been corrected in the M and M section.

3- The GTP Rho pull down experiment not described properly in methods section.

Response: Has now been explained in detail.

4- Was the cyclic AMP assay performed 2-4 hrs after serum starvation or after 24 hrs of starvation?

Response: This has now been corrected in the M and M section regarding cAMP assay.

5- Page 15, correct that GPR56 protein is not expressed in HEK cells

Response: This has now been corrected and included on page 15.

6- Fig 3F-Is the control scrambled siRNA?

Response: The results were similar in both LNCaP cells wildtype and that transfected with scrambled GPR56 siRNA. In the figure results for LNCaP cells transfected using scrambled GPR56 has been included.

7- Rephrase “androgen transcription” to AR downstream signaling, authors are not looking into AR mRNA levels.

Response: Has now been rephrased.

8- Fig4A-PKAsiRNA set is still showing response to ISO.

Response: The experiment with Isoproterenol has been repeated and the results of PKA activity have been revised. 

9- Fig 7- provide a western blotting with N and C terminal GPR56 antibodies.

Response: The Western Blot experiment has now been performed using GPR56 N-terminal antibody (C-terminal antibody was not available at short notice) and the results have been modified accordingly in text and included as Fig. 8A.

---

## [Decision Letter · Decision Letter 1]

2 Apr 2020

PONE-D-19-31997R1

Activation of GPR56, a novel adhesion GPCR, is necessary for nuclear androgen receptor signaling in prostate cells

PLOS ONE

Dear Dr Bagchi,

Thank you for submitting your manuscript to PLOS ONE. After careful consideration, we feel that it has merit but does not fully meet PLOS ONE’s publication criteria as it currently stands. Therefore, we invite you to submit a revised version of the manuscript that addresses the points raised during the review process.

We would appreciate receiving your revised manuscript by May 17 2020 11:59PM. To enhance the reproducibility of your results, we recommend that if applicable you deposit your laboratory protocols in protocols.io, where a protocol can be assigned its own identifier (DOI) such that it can be cited independently in the future. For instructions see: http://journals.plos.org/plosone/s/submission-guidelines#loc-laboratory-protocols

We look forward to receiving your revised manuscript.

Kind regards,

M. Saleem

Academic Editor

PLOS ONE

Reviewers' comments:

Reviewer's Responses to Questions

**Comments to the Author**

1. If the authors have adequately addressed your comments raised in a previous round of review and you feel that this manuscript is now acceptable for publication, you may indicate that here to bypass the “Comments to the Author” section, enter your conflict of interest statement in the “Confidential to Editor” section, and submit your "Accept" recommendation.

Reviewer #1: All comments have been addressed

Reviewer #2: (No Response)

2. Is the manuscript technically sound, and do the data support the conclusions?

Reviewer #1: Yes

Reviewer #2: Yes

3. Has the statistical analysis been performed appropriately and rigorously? 

Reviewer #1: Yes

Reviewer #2: Yes

4. Have the authors made all data underlying the findings in their manuscript fully available?

Reviewer #1: Yes

Reviewer #2: Yes

5. Is the manuscript presented in an intelligible fashion and written in standard English?

Reviewer #1: Yes

Reviewer #2: Yes

6. Review Comments to the Author

Reviewer #1: (No Response)

Reviewer #2: Authors have addressed most of the comments; I recommend a minor revision for figure 6C. The authors have used GAPDH for loading control instead; they need to use lamin or PCNA as nuclear marker and GAPDH/tubulin as cytoplasmic marker. The presence of GAPDH in nuclear fraction indicates that the fractions are not properly separated.

7. PLOS authors have the option to publish the peer review history of their article (what does this mean?). If published, this will include your full peer review and any attached files.

Reviewer #1: No

Reviewer #2: No

---

## [Author Response · Author response to Decision Letter 1]

4 Jul 2020

Dear Editor,

We thank you for the review of the article. We are happy to see that the reviewers were satisfied on almost all accounts.

As per the request of Reviewer 2, point 6, a fresh Western blot experiment has been performed, where presence of androgen receptor (AR) in the cytoplasmic and nuclear fraction was probed. As control we have probed the blot using both GAPDH (cytoplasmic marker) and Histone H3B (nuclear marker). While GAPDH expression is observed in cytoplasmic fractions only, Histone H3B expression was restricted to the nucleus.

Accordingly, changes have been made in the text and the figure has been replaced. In addition, the name of Mr. Sudhir Kumar has been added to the list of authors as he helped us in performing the last experiment. Also, the Histone H3B antibody was a kind gift from Dr. Rakesh Saxena whose name has now been added in acknowledgement and Materials and Methods.

Hope the manuscript is acceptable to you in its current form.

Thanks, and regards,

Gargi Bagchi

---

## [Editor Report · Decision Letter 2]

10 Jul 2020

Activation of GPR56, a novel adhesion GPCR, is necessary for nuclear androgen receptor signaling in prostate cells

PONE-D-19-31997R2

Dear Dr. Gargi Bagchi

We’re pleased to inform you that your manuscript has been judged scientifically suitable for publication and will be formally accepted for publication once it meets all outstanding technical requirements.

Kind regards,

Mohammad Saleem

Academic Editor-PLOS ONE, 

Masonic Cancer Center, University of Minnesota-Minneapolis

---

## [Editor Report · Acceptance letter]

6 Aug 2020

PONE-D-19-31997R2 

 Activation of GPR56, a novel adhesion GPCR, is necessary for nuclear androgen receptor signaling in prostate cells 

Dear Dr. Bagchi:

I'm pleased to inform you that your manuscript has been deemed suitable for publication in PLOS ONE. Congratulations! Your manuscript is now with our production department. 

Kind regards, 

on behalf of

Dr. MOHAMMAD Saleem 

Academic Editor

PLOS ONE